# Optimizing cross-domain transfer for universal machine learning interatomic potentials

Jaesun Kim [1,6], Jinmu You [1,6], Yutack Park[1], Yunsung Lim[2], Yujin Kang [1], Jisu Kim [1], Haekwan Jeon [1], Suyeon Ju[1], Deokgi Hong[1], Seung Yul Lee[3], Saerom Choi [1,4], Yongdeok Kim [4], Jae W. Lee [3] & Seungwu Han [1,2,5] ✉

Accurate yet transferable machine-learning interatomic potentials are essential for accelerating materials and chemical discovery. However, many existing universal models are overfitted to narrow chemical spaces or computational protocols, limiting their reliability across diverse chemical and functional domains. Here, we introduce a transferable multi-domain training strategy that jointly optimizes parameters through selective regularization, coupled with a domain-bridging set that aligns potential-energy surfaces across datasets. Systematic ablation experiments show that suggested strategies synergistically enhance out-of-distribution generalization while preserving in-domain fidelity. Based on our observation, we train SevenNet-Omni on 15 open datasets spanning molecules, crystals, and surfaces. Our model achieves state-of-the-art accuracy in cross-domain benchmarks, reaching chemical accuracy in various scenarios including adsorption-energy in catalytic surfaces and metal–organic frameworks. SevenNet-Omni also accurately reproduces high-fidelity properties by effectively transferring knowledge learned from larger, lower-accuracy databases. This framework offers a scalable route toward universal, transferable models that bridge quantum-mechanical fidelities and chemical domains.

By enabling large-scale atomistic simulations with accuracy approaching that of ab initio methods such as density functional theory (DFT), machine-learning interatomic potentials (MLIPs) have significantly expanded the scope of computational materials science[1–5]. Recently, pretrained universal MLIPs (uMLIPs) have attracted considerable attention because they can bypass the time-consuming creation of ab initio datasets for model training[6–16]. The large databases and deep neural-network architectures behind these models enable reasonable accuracy outside the training domain[16,17]. Currently, a multitude of large-scale ab initio databases are publicly available for training uMLIPs, covering most material classes[7,18–32]. Representative databases are shown in Fig. 1a, which indicates that the databases usually focus on specific chemical domains, reflecting distinct community interests. Each database is obtained with a unique set of computational protocols such as exchange-correlation (XC) functional, program, and ionic potentials.

Many current uMLIPs were trained on datasets from specific chemical domains (e.g., inorganic crystals[13–16] or organic molecules[26,33]), limiting their accuracy in untrained material spaces. As materials engineering advances, the demand for realistic simulations

[1]Department of Materials Science and Engineering, Seoul National University, Seoul, Republic of Korea. [2]Research Institute of Advanced Materials, Seoul National University, Seoul, Republic of Korea. [3]Department of Computer Science and Engineering, Seoul National University, Seoul, Republic of Korea. [4]AI Center, Samsung Electronics, Suwon, Republic of Korea. [5]Center for AI and Natural Sciences, Korea Institute of Advanced Study, Seoul, Republic of Korea. [6]These authors contributed equally: Jaesun Kim, Jinmu You. ✉e-mail: hansw@snu.ac.kr

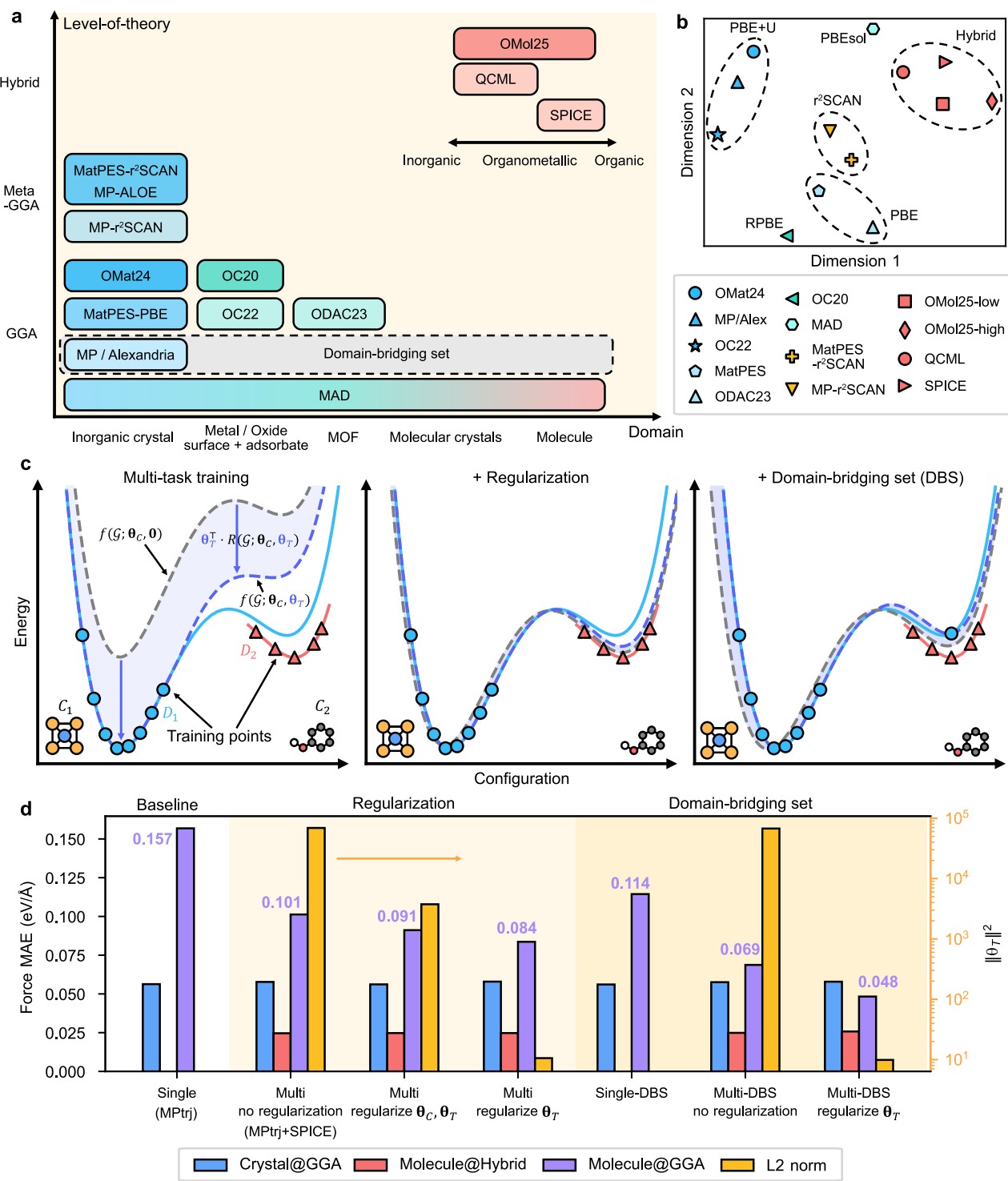

**Fig. 1 | Schematic overview of multi-domain training strategy. a** Training sets used in 7net-Omni, categorized by material domain and level of quantum-mechanical theory. Darker blocks correspond to a higher fraction of high-energy, far-from-equilibrium structures. **b** Clustering of task-embedding vectors projected using principal component analysis. **c** Schematic illustration of potential energy curves obtained from multi-task training. Blue and red solid lines represent the ground-truth PES from the ab initio methods utilized in $D_1$ and $D_2$, respectively, while markers indicate training points from each corresponding database. Dashed lines show PESs predicted by multi-task MLIPs: gray indicates the shared PES, while blue represents the task-specific PES trained on $D_1$. **d** Performance of multi-task MLIPs trained on multi-domain datasets. Blue, red, and purple bars denote force errors for the corresponding material domains and fidelity, while the yellow bar indicates the L2 norm of the task-specific parameters. Source data are provided as a Source Data file.

that span multiple chemical domains is rapidly increasing. Examples include catalytic reactions on metal surfaces in aqueous environments[34,35]; atomic layer deposition (ALD) in semiconductor processing[36]; the formation and evolution of the solid-electrolyte interphase[37–39]. These multi-domain simulations require uMLIPs that can deliver consistent accuracy across chemical domains by being trained over corresponding databases. However, this poses significant challenges in model training because each database is constructed

with different computational protocols, which lead to distinct potential energy surfaces (PESs).

One strategy to incorporate heterogeneous datasets is to align the energy references by shifting and scaling the energies in each database separately[40,41]. However, simply combining heterogeneous databases, even with element-dependent linear transformations, can introduce substantial noise. For example, in Supplementary Fig. 1, the PES of molecular rotation and the water dimer, calculated using various XC functionals, exhibit markedly different profiles. These non-linear discrepancies imply that PES alignment cannot be achieved solely through linear transformations. Hence, a multi-task training framework is required, in which each database is treated as a distinct task to preserve its intrinsic PES characteristics while enabling shared representation learning across domains. So far, only a few uMLIPs such as DPA-3.1[9] and UMA[8], are trained with the diverse databases concurrently within the multi-task framework.

Because uMLIPs must reproduce energies consistent with a chosen set of computational parameters (e.g., XC functional or basis set), transfer learning to bridge differences across computational settings and chemical domains is essential for training multi-task uMLIPs. However, most prior work, including widely used benchmarks, has evaluated multi-task uMLIPs within single-domain scenarios rather than in complex multi-domain applications[8,9,42]; consequently, efficient methods for maximizing cross-domain knowledge transfer remain underexplored.

In this work, we introduce a multi-domain training strategy that optimizes knowledge transfer across heterogeneous datasets by (i) regularizing task-specific parameters and (ii) incorporating small cross-domain bridging sets. Using this strategy, we build an equivariant uMLIP, SevenNet-Omni (7net-Omni hereafter), based on the multi-fidelity SevenNet architecture[43], trained on 15 open databases and 250 million structures released by mid-2025. Extensive multi-domain benchmarks show that 7net-Omni consistently outperforms current frontier uMLIPs.

## Results

### Training strategies for cross-domain generalization

In this study, we address the training of MLIPs using multiple ab initio databases that differ both in the types of atomic configurations (e.g., inorganic versus organic systems) and in the underlying computational approaches (e.g., semilocal versus hybrid functionals). To construct a single MLIP capable of accommodating databases generated under heterogeneous computational settings, we employ a multi-task MLIP framework, where each task corresponds to a different database. (In our original paper, we referred to this approach as *multi-fidelity*[43]. Since the present datasets often share the same level of fidelity, such as Perdew-Burke-Ernzerhof (PBE)[44] and revised PBE (RPBE)[45] within the generalized gradient approximation (GGA), we now use the term *multi-task* for the same methodological approach.) Within this framework, the model parameters are divided into two categories: (i) shared parameters ($\boldsymbol{\theta}_C$) that are universally applied across all databases, and (ii) task-specific parameters ($\boldsymbol{\theta}_T$) that are optimized exclusively for the database corresponding to task $T$. Formally,

$$\text{DFT}_T(\mathcal{G}) \approx f(\mathcal{G}; \boldsymbol{\theta}_C, \boldsymbol{\theta}_T), \tag{1}$$

where $\text{DFT}_T$ denotes the reference label obtained within DFT corresponding to task $T$, $f$ represents the MLIP model, and $\mathcal{G}$ indicates atomic configuration. Through the shared parameter set $\boldsymbol{\theta}_C$, knowledge gained from one database is transferred to the others.

For the continuity of inferred force from MLIP models, one usually avoids non-smooth activation functions such as Rectified Linear Unit (ReLU)[46]. This guarantees model become at least $C^1$ function for parameter space. Therefore, one can apply Taylor's theorem to Eq. (1)

with respect to $\boldsymbol{\theta}_T$:

$$
\begin{aligned}
f(\mathcal{G}; \boldsymbol{\theta}_C, \boldsymbol{\theta}_T) &= f(\mathcal{G}; \boldsymbol{\theta}_C, \mathbf{0}) + \boldsymbol{\theta}_T^\top \cdot \nabla_{\boldsymbol{\theta}_T} f(\mathcal{G}; \boldsymbol{\theta}_C, \mathbf{0}) \\
&\quad + \boldsymbol{\theta}_T^\top \cdot R_1(\mathcal{G}; \boldsymbol{\theta}_C, \boldsymbol{\theta}_T) \\
&= f(\mathcal{G}; \boldsymbol{\theta}_C, \mathbf{0}) + \boldsymbol{\theta}_T^\top \cdot R(\mathcal{G}; \boldsymbol{\theta}_C, \boldsymbol{\theta}_T),
\end{aligned}
\tag{2}
$$

where

$$R(\mathcal{G}; \boldsymbol{\theta}_C, \boldsymbol{\theta}_T) := \int_0^1 \nabla_{\boldsymbol{\theta}_T} f(\mathcal{G}; \boldsymbol{\theta}_C, t\,\boldsymbol{\theta}_T)\, dt \tag{3}$$

can be obtained by integration since $\nabla_{\boldsymbol{\theta}_T} f(\mathcal{G}; \boldsymbol{\theta}_C, t\,\boldsymbol{\theta}_T)$ is continuous thus Riemann integrable. The right-hand side of Eq. (2) thus separates the contributions that do and do not depend on $\boldsymbol{\theta}_T$. Namely, $f(\mathcal{G}; \boldsymbol{\theta}_C, \mathbf{0})$ depends solely on the shared parameters $\boldsymbol{\theta}_C$ and is therefore referred to as the common PES. The second term represents the task-specific contribution.

The left panel of Fig. 1c schematically illustrates the training outcomes and the role of each term in Eq. (2) when two databases ($D_1$ and $D_2$) with distinct atomic structures ($C_1$ and $C_2$) and computational methods are employed. The blue and red solid curves represent the ground-truth PESs of the respective ab initio methods, with markers along each curve indicating the training data points. We focus on the task $T$ associated with $D_1$. The blue dashed curve corresponds to the MLIP prediction incorporating $\boldsymbol{\theta}_T$, fitted to the blue data points. The gray dashed curve denotes the common PES [$f(\mathcal{G}; \boldsymbol{\theta}_C, \mathbf{0})$ in Eq. (2)], while the shaded blue region highlights the task-specific contribution.

Since the resulting MLIP PES is expressed as the sum of the common PES and the task-specific contribution, multiple combinations of $\boldsymbol{\theta}_C$ and $\boldsymbol{\theta}_T$ can yield comparable training losses. As depicted in the left panel of Fig. 1c, when the task-specific contribution dominates, the MLIP prediction (blue dashed curve) deviates significantly from the ground truth (blue solid curve) at $C_2$, which exists only in $D_2$. This suboptimal transfer occurs because the common PES is inaccurate, while the task-specific contribution is overfitted to a single domain ($C_1$) and therefore performs poorly in out-of-distribution regions.

To address this issue, we introduce regularization on $\boldsymbol{\theta}_T$. Motivated by Eq. (2), the task-specific contribution term can be suppressed by regularizing the size of $\boldsymbol{\theta}_T$, since the absolute value of the remainder in Eq. (3) has an upper bound. By penalizing large values of $\boldsymbol{\theta}_T$, regularization encourages the model to rely more heavily on the common PES, ensuring that the shared representation captures the essential bonding characteristics across both databases. This is illustrated schematically in the central panel of Fig. 1c, where the common PES now remains close to the ground-truth PESs of both tasks, thereby enhancing knowledge transfer between the two databases.

While the regularization of $\boldsymbol{\theta}_T$ is effective in improving transfer learning, it tends to fit the PES to the ground truth of $D_2$ in the $C_2$ domain. This limitation arises from the complete absence of $D_1$ at $C_2$. To address this, we introduce a domain-bridging set (DBS), which augments $D_1$ by evaluating a subset of $D_2$ with the ab initio method used for $D_1$ (see the right panel of Fig. 1c). Incorporating a DBS requires only marginal computational effort (see below) yet yields substantial improvements in the accuracy of task $T$ at $C_2$.

To demonstrate the effectiveness of the proposed approach, we employ the MPtrj[7] and SPICE[27] databases for MLIP training. The MPtrj database, representative of inorganic bulk systems, is generated using the PBE functional, whereas the SPICE database, curated for organic molecules, is based on hybrid functional of $\omega$B97M[47]. For multi-task modeling, we adopt the SevenNet-MF architecture[43] and evaluate model performance by computing the force mean absolute error (MAE) relative to ab initio reference results.

The evaluation results are presented in Fig. 1d. All evaluations are performed using the validation sets that are not utilized for training. As

**Table 1 | List of the databases used to train 7net-Omni**

| Database abbreviation | Task (Channel) | Number of structures/10³ | Domain | XC functional | Oversampling | Reference |
|---|---|---|---|---|---|---|
| MPtrj | mpa | 1580 | Inorganic crystal | PBE(+$U$) | 15 | 7 |
| Alex[1] | mpa | 12,069 | Inorganic crystal | PBE(+$U$) | 4 | 18–20 |
| DBS | mpa | 121 | General | PBE(+$U$) | 15 | - |
| OMat24 | omat24 | 101,901 | Inorganic crystal | PBE(+$U$) | 1 | 21 |
| MatPES | matpes | 412 | Inorganic crystal | PBE | 40 | 22 |
| OC20[2] | oc20 | 30,757 | Catalyst (metal) | RPBE | 2 | 23 |
| OC22 | oc22 | 8210 | Catalyst (oxide) | PBE(+$U$) | 6 | 24 |
| ODAC23 | odac23 | 4082 | MOF | PBE-D3 | 1 | 25 |
| OMol25 (low)[3] | omol25 | 60,852 | Molecule | $\omega$B97M-V | 1 (5)[3] | 26 |
| OMol25 (high)[3] | omol25_high | 1390 | Molecule | $\omega$B97M-V | 5 | 26 |
| SPICE[4] | spice | 1738 | Molecule | $\omega$B97M | 5 | 27 |
| QCML | qcml | 18,301 | Molecule | PBE0[118] | 1 | 28 |
| MAD | mad | 86 | General | PBEsol[119] | 100 | 29 |
| MP-r²SCAN | mp_r2scan | 50 | Inorganic crystal | r²SCAN | 40 | 93,120 |
| MatPES-r²SCAN | matpes_r2scan | 368 | Inorganic crystal | r²SCAN | 40 | 22 |
| MP-ALOE | matpes_r2scan | 864 | Inorganic crystal | r²SCAN | 15 | 30 |

Each entry provides the database abbreviation, the corresponding task name, the number of structures included in training, the chemical domain, the XC functional used in the calculations, the oversampling factor applied during training, and the corresponding literature reference. Charged structures in SPICE, OMol25 and QCML are excluded. Databases with identical computational protocols are grouped under the same task. The total number of structures is 242 million.

[1] A subset of the Alexandria dataset (sAlex) is included for 3D configurations[21], and 2D and 1D configurations are also incorporated.

[2] The OC20 database consists of relaxation trajectories, rattled structures, and ab initio molecular dynamics configurations. For the relaxation trajectories, we employ the OC20M split provided in OC20, while for the rattled and MD structures, we use subsampled datasets. See Methods section for detailed subsampling criteria.

[3] We split the OMol25 database, which contains various spin states, into 'low-spin' and 'high-spin' configurations, treating these two classes as separate tasks. The organometallic complex structures in the low-spin category are oversampled five times to improve training. See the Methods section for the splitting criteria.

[4] We use energies and forces calculated without the D3 dispersion correction.

a baseline, we trained a single-task model using only the MPtrj database (left region), which exhibits large errors for molecular structures evaluated at the PBE level (denoted as Molecule@GGA). We then examined the impact of different regularization strategies in SevenNet-MF training, comparing a multi-task MLIP without regularization, a model trained with regularization applied to both $\boldsymbol{\theta}_C$ and $\boldsymbol{\theta}_T$, and a model trained with regularization applied only to $\boldsymbol{\theta}_T$. As shown in the middle region of Fig. 1d, all models achieve comparable force accuracy within their respective training domains and computational settings (Crystal@GGA and Molecule@Hybrid), regardless of the regularization scheme. However, the force MAE differs significantly when evaluated on out-of-domain organic molecular configurations at the PBE level (Molecule@GGA), while all multi-task MLIPs outperform the single-task baseline. In particular, selective regularization of $\boldsymbol{\theta}_T$ yields greater improvements than conventional regularization applied to all model parameters, underscoring its advantage for optimizing cross-database transferability. The right axis shows that model accuracy systematically improves as the L2 norm of $\boldsymbol{\theta}_T$ decreases, indicating that effective knowledge transfer is facilitated by suppressing task-specific contributions.

Next, we extend our evaluation to scenarios where a DBS is available (right region of Fig. 1d). For the DBS, we randomly subsample 0.1% of atomic configurations from the SPICE database and recompute them using the same computational settings as those used for MPtrj. As shown in Single-DBS, accuracy remains insufficient when training with only MPtrj and DBS. In contrast, supplementing with the full SPICE database through multi-task training (Multi-DBS) yields a substantial performance improvement. Furthermore, combining DBS with selective task regularization produces a synergistic effect, achieving the best overall accuracy among all training strategies.

We further examine the effects of regularization and DBS on each term in Eq. (2) by analyzing the PES associated with the binding of two water molecules using the multi-task models shown in Fig. 1d. For each model, two task-specific PES are obtained by selecting $\boldsymbol{\theta}_T = \boldsymbol{\theta}_{\mathrm{MPtrj}}$ and

$\boldsymbol{\theta}_T = \boldsymbol{\theta}_{\mathrm{SPICE}}$, where $\boldsymbol{\theta}_{\mathrm{MPtrj}}$ and $\boldsymbol{\theta}_{\mathrm{SPICE}}$ denote parameters trained on the MPtrj and SPICE databases, respectively. In addition, the common PES is evaluated by explicitly setting $\boldsymbol{\theta}_T = \mathbf{0}$ during inference.

The resulting PES for each model evaluated on molecular input structures are shown in Supplementary Fig. 2. The PES inferred using $\boldsymbol{\theta}_{\mathrm{SPICE}}$ exhibits good agreement with the $\omega$B97M reference across all training methods. In contrast, the agreement between PBE and the PES obtained with $\boldsymbol{\theta}_{\mathrm{MPtrj}}$ varies substantially, consistent with the trends shown in Fig. 1d. Comparison of Supplementary Fig. 2a, b indicates that regularization renders the common PES more physically meaningful, yielding a more generalizable representation of $\boldsymbol{\theta}_C$ and improved agreement with PBE. By contrast, comparison of Supplementary Fig. 2a, c shows that DBS enhances accuracy by improving the task-specific contribution associated with $\boldsymbol{\theta}_{\mathrm{MPtrj}}$, while the common PES remains suboptimal. Together, these results demonstrate that the proposed training strategies operate as intended and clarify the synergistic effect of regularization and DBS, as illustrated in Supplementary Fig. 2d.

Building on these observations, we developed a uMLIP, 7net-Omni, by concurrently training the SevenNet-MF architecture on 15 publicly available databases (covering 13 different computational protocols) with selective task regularization and DBS. The training set spans a diverse collection of ab initio databases encompassing molecular, crystalline, metal−organic framework (MOF), and surface systems, as schematically illustrated in Fig. 1a. The specific databases, along with their chemical domains and XC functionals, are listed in Table 1. The present uMLIP concerns the charge neural system, so charged structures in SPICE, OMol25, and QCML are excluded. Each database is assigned to a distinct task, but databases with identical computational protocols are grouped under the same task. We also use the term *channel* to refer to task-specific inference.

To construct DBS, we adopted the computational settings used in the MPtrj database and performed single-point calculations on approximately 0.1% of structures from six representative databases

(see Methods for details of the sampling criterion). The numbers of sampled structures from each database are summarized in Supplementary Table 1.

To efficiently train on heterogeneous datasets, we adopted a curriculum learning strategy. Specifically, the model was first trained on crystal databases such as MPtrj, sAlex, and OMat24 to establish a foundation of chemical knowledge. The training was then extended to molecular systems by introducing OMol25 into the dataset. Finally, the model was trained on the whole database listed in Table 1. This curriculum-based training process enables the model to effectively capture complex chemical environments, such as adsorbate-slab systems, by utilizing prior knowledge established for crystalline and molecular systems. When a new database is introduced as an additional task, the corresponding PES is adapted by initializing the energy-shift parameters via linear regression and making them trainable.

This approach resulted in a more stable learning process compared to joint training on all databases from scratch, which we found to suffer from unstable optimization and poor initial convergence in a highly heterogeneous distribution of the training set. Similar training procedures have previously been adopted to integrate heterogeneous databases into a single MLIP model[41]. During this curriculum learning, the predictive accuracy on crystalline databases showed no degradation (Supplementary Fig. 3), indicating that progressively expanding the training dataset did not induce noticeable catastrophic forgetting on previously trained domains. We note that the specific ordering of the curriculum was not systematically optimized, leaving room for further improvement in the choice of curriculum sequence. Further details of dataset composition, sampling ratios, and training procedures are provided in the Methods section.

To examine whether the model encodes computational methods in an explainable way, we analyzed the trained parameters of 7net-Omni associated with task embeddings. Since the base architecture, SevenNet-MF, employs one-hot encoding of each task in each self-interaction layer[43], the corresponding $\theta_T$ parameters in each self-interaction layer themselves serve as task-embedding vectors. Specifically, we used the $\theta_T$ from the first self-interaction layer of SevenNet-Omni. The latent vectors were visualized using principal component analysis (PCA) in Fig. 1b. We find that the task-dependent weights form distinct clusters that primarily correlate with the choice of XC functional and the application of a Hubbard $U$ correction[48]. For example, molecular databases employing hybrid functionals group together in a similar region. This indicates that the model captures both similarities and discrepancies across different levels of theory, thereby enhancing the generalizability of the multi-task framework. Comparable analyses have been reported in the context of multi-task property prediction using the multiXC-QM9 dataset[49,50]. However, unlike the present observation, it produced mixed results; for instance, embeddings generated with the same XC functional but different basis sets exhibited low similarity. This difference may stem from the use of selective task regularization in 7net-Omni, which allows the model to better recognize cross-theory similarities.

## Frontier uMLIPs

Throughout this paper, we compare the performance of 7net-Omni with other top-performing multi-task uMLIPs, namely UMA (UMA-m-1p1)[8] and DPA (DPA-3.1-OpenLAM)[9], as well as single-task uMLIPs: eSEN (eSEN-30M-OAM, eSEN-30M-OMAT)[12], ORB (orb-v3-conservative-inf-mpa, orb-v3-conservative-inf-omat)[13], 7net-ompa (SevenNet-MF-ompa), NequIP (Nequip-OAM-L)[14,51], GRACE (GRACE-2L-OAM-L, GRACE-2L-OMAT-L)[15], and MACE (MACE-mpa-0-medium, MACE-omat-0-medium)[52,53]. The specific model names tested in this work are indicated in parentheses and these correspond to the most accurate variants within each model family, as far as we can determine. We note that all the models are equivariant graph neural networks except for DPA-3.1 which utilizes invariant features. The architectural

categorization of ORB remains ambiguous. While ORB employs interatomic displacement vectors as equivariant input features[54], its use of multi-layer perceptrons in the model architecture means that strict roto-equivariance and exact energy invariance are not enforced[13]. Instead, ORB adopts the *equigrad* loss, which penalizes large energy discrepancies between rotated configurations rather than guaranteeing strict equivariance by construction[13]. All the models are conservative, producing atomic forces as gradients of the energies.

We classify UMA and DPA as multi-task uMLIPs, as they are trained concurrently on databases spanning diverse material classes, including inorganic bulk, surfaces, and organic molecules. UMA is trained on OMat24, OC20, ODAC25[31], OMC25[32], and OMol25, whereas DPA is trained on OpenLAM-v1, which aggregates 31 databases, including large-scale datasets such as OMat24, MPtrj, OC20, and SPICE, as well as several domain-specific databases. Architecturally, UMA incorporates additional embedding layers that encode information about the DFT task, charge, and spin, which are appended to the node features at each graph convolution layer. By contrast, DPA employs a one-hot encoding scheme at the final multilayer perceptron stage to distinguish training databases when predicting atomic energies. In terms of regularization, UMA uses the AdamW optimizer with a small weight decay applied to all model parameters, while DPA adopts the Adam optimizer without regularization. Thus, neither UMA nor DPA introduced the task-specific regularization proposed in this work.

Single-task uMLIP models, including eSEN, ORB, NequIP, GRACE, and MACE, typically have two variants depending on the choice of database: OMat24 or MPtrj/Alexandria, which were generated using slightly different pseudopotential sets[21]. In most cases, the models are first pretrained on the OMat24 database to capture broad, high-energy configurations (e.g., eSEN-30M-OMAT). These OMat24-pretrained uMLIPs are then fine-tuned on the MPtrj and Alexandria databases (e.g., eSEN-30M-OAM).

Since the present work focuses mainly on PBE-based benchmarks, we concentrate on the tasks for each uMLIP in the OMat24 and MPtrj/Alexandria databases. For 7net-Omni, because the DBS is constructed using the computational setup of MPtrj, the mpa channel is generally the most accurate, although this can vary slightly depending on the examples considered, as discussed in later sections. We also consider the matpes channel of 7net-Omni, since its exclusion of the Hubbard $U$ term proves useful for applications including transition metals (see Cross-domain or cross-functional scenarios and Metallic surfaces sections). Although 7net-ompa was trained in a multi-task architecture (concurrently on MPtrj, sAlex, and OMat24), we classify it as a single-task model because its training data consist solely of inorganic crystals. Since 7net-ompa share the same architecture and hyperparameters with 7net-Omni, the performance gain of 7net-Omni over 7net-ompa can be attributed to the additional databases and the multi-domain training strategy proposed here.

For notational simplicity, we henceforth denote multi-task models in the form model.task (e.g., UMA.omc, DPA.spice) and single-task models in the form model[dataset] (e.g., eSEN[oam] for eSEN-30M-OAM, MACE[omat] for MACE-omat-0-medium).

We emphasize that in the following benchmark tests, all physical quantities are computed self-consistently within the given uMLIP or ab initio method, independent of results from other approaches. For example, when estimating reaction energies between two states, each state is obtained by fully relaxing the structure within the model itself. For benchmarks employing van der Waals interactions at the D3 level, we add the D3 correction separately if the task was not trained with D3 included (e.g., 7net-Omni.omat24). If the database already incorporates van der Waals interactions, we use the task results without further modification (e.g., 7net-Omni.omol25 and UMA.omc). We add a cautionary remark that the performance improvements reported in this work primarily reflect better agreement with the PBE functional, which does not necessarily imply closer correspondence to physical reality

due to the inherent approximations of GGA functionals and DFT. When comparisons are made with higher levels of theory, such as the meta-GGA r$^2$SCAN functional, or experimental data, we explicitly note the reference methods.

## Single-domain applications

Before conducting extensive multi-domain benchmarking, we first examine in-domain cases where both the material type and XC functional are represented in the training set. One such benchmark is Matbench Discovery[42], which reports F1, $\kappa_{SRME}$, and RMSD, each probing a different facet of reliability. The F1 score (harmonic mean of precision and recall) measures the accuracy of crystalline energy ranking; $\kappa_{SRME}$ is the symmetric relative mean error (SRME) for lattice thermal conductivity ($\kappa$), reflecting the fidelity of curvature at the PES minimum; and RMSD quantifies structural deviations after relaxation relative to DFT. A combined performance score (CPS) summarizes these metrics. 7net-Omni.mpa achieves F1 = 0.889, $\kappa_{SRME}$ = 0.265, and RMSD = 0.0639, yielding CPS = 0.849, which is slightly higher than 7net-ompa.mpa (0.845). Notably, $\kappa_{SRME}$ for 7net-Omni.omat24 and 7net-Omni.matpes are 0.253 and 0.243, respectively, lower than for 7net-Omni.mpa, which can be attributed to the inclusion of high-energy configurations in OMat24 and MatPES.

To extend beyond tests on pure crystals, we compute grain boundary (GB) energies for 327 configurations across 58 elemental metals[55], covering a range of misorientation angles as well as tilt and twist boundaries. We exclude 30 GBs with non-orthogonal planes relative to the z-direction[15]. The GB energy ($\gamma_{GB}$) is computed following the definition in ref. 55, and performance is quantified as the SRME of $\gamma_{GB}$. Supplementary Fig. 4a summarizes the performance of 7net-Omni compared with other leading uMLIPs. Parity plots for GB energy benchmark are also illustrated in Supplementary Fig. 5. All models exhibit comparable performance, but 7net-Omni.mpa performs slightly better.

As another non-crystalline benchmark, we compute defect binding energies in steels. Previous DFT studies have examined interactions among carbon interstitials and vacancies, as well as interactions between transition-metal solute atoms[56,57], which have been adopted as a benchmark for uMLIPs[58]. In this benchmark, we compute binding energies between carbon and vacancy, as well as between transition-metal solutes (Ti, V, Cr, Mn, Co, Ni, Cu, Nb, and Mo). For all relaxations of defective structures, both atomic positions and lattice parameters are allowed to vary, while the cell shape is constrained to remain orthogonal. Supplementary Fig. 4b, c compare the accuracy of uMLIPs for defect binding energies, as well as their parity plots can be found in Supplementary Figs. 6, 7. Overall, all models achieve reasonable accuracy, with no single model showing a clear advantage.

While PBE functionals remain the standard approach in inorganic solid-state simulations, hybrid functionals are more widely adopted in molecular modeling because they provide more accurate descriptions of reaction barrier heights, a task where semilocal functionals often struggle[39]. To evaluate the hybrid-functional fidelity of multi-task models, we consider small-molecule torsion barriers as a benchmark for conformational PES descriptions. Two pharmaceutically relevant datasets, the biaryl set and TorsionNet500[59,60], were recently recomputed at the $\omega$B97M-D3 level of theory[47,61], yielding torsion barriers. From these, we select all 88 molecules in the biaryl set and 100 molecules sampled from TorsionNet500. Torsional barrier heights are defined as the energy difference between the minimum and maximum points along the torsional PES[59].

In Supplementary Fig. 4d, we benchmark several uMLIPs, including 7net-Omni.spice, 7net-Omni.omol25, DPA.spice, and UMA.omol25 as multi-task models, alongside single-task models for organic molecules, such as MACE-OFF24-medium trained on the SPICE dataset[27,33] (denoted MACE[spice]) and eSEN-sm-conserving trained on the OMol25 dataset[26] (denoted eSEN[omol]). (See Supplementary Fig. 8 for

parity plots.) Although the dispersion corrections differ between the functionals used in SPICE ($\omega$B97M-D3) and OMol25 ($\omega$B97M-V[62]), the results are compared on equal footing. As shown in Supplementary Fig. 4d, all models achieve high accuracy for torsional barriers, with MAEs well below the chemical accuracy threshold of 1 kcal/mol (gray dashed line in Supplementary Fig. 4d).

## Cross-domain or cross-functional scenarios

The main motivation of this work is to develop a uMLIP that can simulate complex material systems in which distinct classes of materials interact. One example is semiconductor processing, such as ALD, where precursor molecules adsorb onto silicon-based substrates. In such applications, semilocal functionals, most commonly the PBE functional, are preferred to describe both molecular and solid-state systems simultaneously[63,64]. Within the current multi-task training framework, this requires effective knowledge transfer from PESs learned with hybrid functionals to those learned with the PBE functional for molecules. To examine this more closely, we first focus on molecular systems, either isolated or crystalline, computed at the PBE level, as shown in Fig. 2a–c. (For parity plots of each benchmark, see Supplementary Figs. 9–11).

First, small-molecule torsion barriers studied in the previous section are selected as a single-molecule benchmark for evaluating the accuracy of conformational PESs in complex molecules. The molecules are reoptimized at the PBE-D3 level to directly compare energy barriers with PBE-fidelity channels. The MAE in torsion barriers is summarized in Fig. 2a, which shows that 7net-Omni achieves significantly higher accuracy than 7net-ompa, benefiting from the incorporation of molecular databases. Notably, 7net-Omni.mpa attains the hybrid-functional accuracy of 7net-Omni.spice reported in the previous section (see the white bullet). This demonstrates successful knowledge transfer across both chemical domains and fidelity. UMA.omc and DPA.mp show similar trends; however, they exhibit large variations in accuracy among channels, whereas 7net-Omni maintains more uniform error levels, attributable to the selective task regularization.

In Supplementary Fig. 9, parity plots for torsion barriers are shown for all models. As the MAE increases, the data points become more scattered, indicating that the errors are more random rather than systematic. This increased randomness makes the predictions less reliable. This trend is consistently observed throughout this work: larger MAE values generally correspond to weaker correlations.

In Fig. 2a, multi-task models on accurate channels outperform most single-task models, underscoring the role of training data for molecules. The exception is eSEN[oam], which achieves accuracy comparable to that of 7net-Omni. The relatively high accuracy of eSEN[oam] on systems involving small molecules is consistently observed throughout this work. The unexpected performance of eSEN[oam] outside its training domain has also been reported in other studies[65,66], although it remains unexplained at present.

Next, we compute the reaction energy ($E_{rxn}$) for organometallic complexes. A total of 53 reactions involving 97 organometallic complexes were collected from the literature and reoptimized at the PBE-D3 level of theory[67,68]. Figure 2b shows the MAE of $E_{rxn}$ over the benchmark set, indicating that 7net-Omni outperforms all other models. Both 7net-Omni and UMA include organometallic complexes in their training sets (OMol25), yet their cross-functional performances differ substantially. Interestingly, 7net-Omni.matpes is more accurate than 7net-Omni.mpa, despite the latter channel incorporating some organometallic complexes through DBS. Further analysis reveals that reactions with large errors in 7net-Omni.mpa predominantly involves Cr, Fe, Co, and Ni centers (see Supplementary Fig. 10), suggesting that this underperformance is related to the application of PBE+$U$ in the MPtrj/sAlex database for partially filled 3$d$ transition metals. A more detailed discussion is provided in Metallic surfaces section.

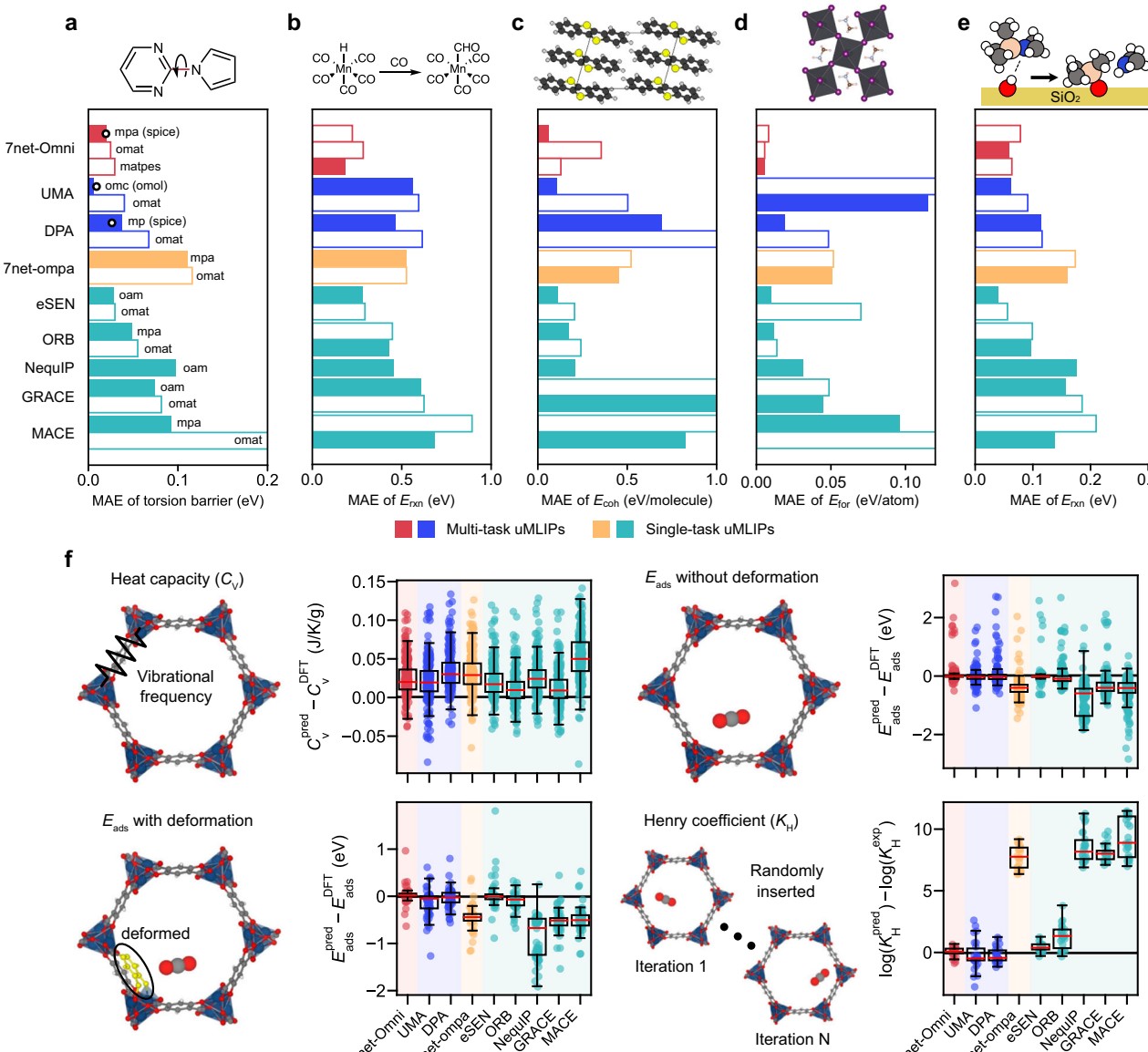

**Fig. 2 | Performance of uMLIPs for cross-domain scenarios. a** MAE in predicting torsional energy barriers. The *y*-axis lists the uMLIP models. For multi-task uMLIPs, the inference channel is indicated to the right of each bar; for single-task uMLIPs, the corresponding training set is shown. White bullets mark the accuracy of the hybrid-functional channel (parentheses) in reference to the *ω*B97M-D3 results. **b** MAE of reaction energy predictions for organometallic complexes. **c** MAE of cohesive energy predictions for organic crystals. **d** MAE of formation energy predictions for hybrid organic-inorganic perovskites. **e** MAE of adsorption energy predictions for molecular inhibitors, computed by energy changes from

physisorbed to chemisorbed states. **f** Error distributions of four benchmark tasks for metal--organic frameworks, represented as box plots. The box and red horizontal line in the box plot show the quartiles and median of the error distribution, respectively, while whiskers represent 1.5 times the inter-quartile range. Individual data points are randomly jittered along the horizontal axis for visual clarity. In (**a**–**f**), all reference DFT data are obtained with PBE-D3. Solid bars in (**a**–**e**) represent the best-performing channel or training database. Individual parity plots are provided in Supplementary Figs. 9–11,14,15. Source data are provided as a Source Data file.

To assess the description of intermolecular interactions, which are critical in organic semiconductors and molecular liquids, we investigate molecular crystals. A total of 86 molecular crystals are considered, comprising 23 from the X23 set[69,70] and 67 from the BMCOS1 benchmark set[71], spanning systems from simple crystals to optoelectronic materials. Four crystals that overlap with the BMCOS1 set are excluded from the X23 subset to avoid redundancy. We compute the cohesive energy ($E_{coh}$), defined as the energy required to separate the crystal into its constituent molecules. The MAE of $E_{coh}$ in Fig. 2c shows that 7net-Omni.mpa outperforms other models, including UMA.omc, which is explicitly trained on molecular crystals. (For parity plots, see Supplementary Fig. 11). Supplementary Figs. 12, 13

further compare errors in the equilibrium volumes of molecular crystals, showing trends consistent with those in Fig. 2c.

Next, we consider the ABX₃-type hybrid organic-inorganic perovskites as an example of multi-domain applications, which have attracted considerable attention in photovoltaics and optoelectronics[72,73]. Although some hybrid perovskites are included in databases such as MPtrj, diverse combinations of A, B, and X remain under-represented. Kim et al.[74] constructed a comprehensive database of hybrid organic-inorganic perovskites, including one of the 16 molecular cations at the A site and a metal of the IV group at the B site. We randomly select 100 hybrid perovskite structures from the database and calculate the theoretical formation energy ($E_{for}$), which

represents the relative energy difference between the perovskite and its components (e.g., metal and molecule) at the PBE-D3 level:

$$E_{for} = E^{ABX_3} - E^{A'} - E^{B} - \frac{3}{2}E^{X_2} - \frac{1}{2}E^{H_2}, \quad (4)$$

where $E^{\alpha}$ is the energy of $\alpha$. In Eq. (4), A′ indicates the neutral organic molecule corresponding to the cation $A^+$ (e.g., $NH_3$ for $NH_4{}^+$), and $E^B$ denotes the total energy of the elemental metal B in its stable crystalline phase.

The benchmark results are shown in Fig. 2d, as well as corresponding parity plots are represented in Supplementary Fig. 14. Across all tested models, 7net-Omni shows the best agreement with DFT. Notably, both UMA.omc and UMA.omat exhibit relatively large errors: the omat channel suffers from inaccurate molecular energies, suggesting limited transferability from the OMol25 database, while the omc channel struggles to identify stable structures of inorganic crystals. In contrast to eSEN[oam], eSEN[omat] shows large errors for certain molecules (e.g., $F_2$, HF, NO, $O_2$), which results in low accuracy in Fig. 2d. This issue is further analyzed later in this section.

In Fig. 2e, we extend our analysis to surface reactions involving the adsorption of organic molecules on dielectric substrates, motivated by thin-film deposition in semiconductor processing. Specifically, we consider Si-centered inhibitor molecules adsorbing on $SiO_2$ and $Si_3N_4$ substrates, which are relevant to area-selective deposition (ASD)[75]. We adopt 78 PBE-D3 reference results for physisorption and chemisorption from ref. 76. For the adsorbates, we examine (N,N-dimethylamino) trimethylsilane (DMATMS) and ethyltrichlorosilane (ETS), while $SiO_2$ and $Si_3N_4$ serve as substrates. The reaction energy ($E_{rxn}$) is defined as the energy difference between the chemisorbed and physisorbed states,

$$E_{rxn} = E^{chem} - E^{phys}. \quad (5)$$

Figure 2e summarizes the model accuracy, showing that 7net-Omni achieves substantial improvements over 7net-ompa. Among the single-task models, eSEN and ORB also perform competitively. (For parity plots, see Supplementary Fig. 15).

Metal-organic frameworks (MOFs), as hybrid organic-inorganic materials, offer vast structural tunability through the nearly limitless combinations of linkers and metal nodes[77]. However, their immense chemical space and large atomic numbers often make DFT-based screening impractical. Recent advances in uMLIPs provide a scalable alternative[78]. To assess the accuracy of 7net-Omni in MOF applications, we conducted four benchmark tasks covering both homogeneous properties (heat capacity) and heterogeneous cases (MOF-guest interactions).

Full data in Supplementary Fig. 16 show that all models performed well in predicting heat capacities[79,80], producing MAEs less than 0.03 J/ K/g, but accuracy in MOF-guest interactions varied widely depending on channels and training databases for some models (UMA, DPA, and eSEN). Therefore, throughout Fig. 2f, we display only accurate variants: 7net-Omni.mpa, UMA.omc, DPA.mp, 7net-ompa.mpa, eSEN[oam], ORB[omat], NequIP, GRACE[omat], and MACE[mpa]. Except for UMA.omc, all models include D3 corrections separately.

Next, we evaluated adsorption of single $CO_2$ and $H_2O$ molecules using the test set from the GoldDAC dataset[81], following established protocols (see upper-right section in Fig. 2f). The adsorption energy, $E_{ads}$, of a molecule X is defined as

$$E_{ads} = E^{MOF + X} - E^{MOF} - E^{X}, \quad (6)$$

where $E^X$ is the energy of the isolated molecule X (X=$CO_2$, $H_2O$). In this task, 7net-Omni.mpa achieved accuracy comparable to eSEN[oam] and close to UMA.odac, which was specifically designed for this type of

evaluation (Supplementary Fig. 16b). Similar trends were observed for predicting experimental $CO_2$ Henry coefficients ($K_H$) from the ODAC25 study[31] (see lower-right section in Fig. 2f), reflecting the general proportionality between adsorption energies ($E_{ads}$) and $K_H$.

In Supplementary Fig. 16b, 7net-Omni.odac23 underperforms in $E_{ads}$ without deformation in the repulsion region, even though this task is directly trained with the database related to MOF-gas interactions. We suspect that this behavior arises from the relatively loose DFT settings used in the ODAC23 dataset, which may have introduced noise[25]. In preliminary tests, replacing ODAC23 with ODAC25 (without updating DBS) significantly reduced the error especially in the repulsive region, as illustrated in Supplementary Fig. 17.

In the lower-left region of Fig. 2f, we further evaluated adsorption with framework deformation using a DFT dataset from prior work[82], again focusing on single-molecule $CO_2$ and $H_2O$ adsorption. In this scenario, 7net-Omni.mpa delivered the highest accuracy, surpassing the practical 0.1 eV MAE threshold for screening (gray dashed line in Supplementary Fig. 16d)[82]. Parity plots for the MOF benchmarks are illustrated in Supplementary Figs. 18–22.

In the above benchmark tests, particularly in Fig. 2c,d,f, we find that the accuracy of isolated molecules plays a crucial role in determining overall model performance. Specifically, for the MOF systems discussed above, the error analysis of $E_{ads}$ in Supplementary Fig. 16e indicates that the deviations mainly originate from energy errors of the adsorbates ($CO_2$ and $H_2O$). This is generalized in Fig. 3a, which presents the distribution of energy discrepancy between uMLIPs and DFT for 26 molecules sampled from the examples in this section. It can be seen that single-task models tend to overestimate molecular energies, except for eSEN[oam]. (The differences in DFT settings between MPtrj and OMat24 change the molecular energies by less than 2 meV/atom). The relatively weaker performance of UMA.omat can be attributed to the small fraction of molecule-containing structures in the OMat24 database itself than MPtrj and Alex (see Supplementary Table 2), as well as insufficient knowledge transfer from molecular databases trained for other inference tasks. Similar limitations of multi-task models on low-dimensional configurations have also been reported in a previous study[66]. Since the whole systems exhibit closer agreement with DFT, the isolated molecules become effectively destabilized in many models. As an illustrative example, we investigate the origin of errors in hybrid perovskite systems (Fig. 2d). The total energy errors of bulk phases ($E^{ABX_3}$ and $E^B$) are consistently smaller than those of molecular species ($E^{A'}$ and $E^{X_2}$), which are commonly overestimated by most uMLIPs, as shown in Supplementary Fig. 23a, b. Consequently, errors associated with molecular systems exhibit stronger correlations and contribute more significantly to the $E_{for}$, consistent with the trend observed in Supplementary Fig. 16e (see Supplementary Fig. 23c).

The spurious destabilization of molecules also significantly distorts the PESs. This is illustrated in Fig. 3b, which plots the PES as a $CO_2$ molecule approaches the corner of a MOF. The destabilization causes molecular overbinding and a stiffening of the PES near equilibrium, manifested as increased curvature. The stiffening is further corroborated in Fig. 3c, which presents parity plots of the $E_{ads}$ values without deformation from Fig. 2f. Predictions from two 7net models, 7net-Omni and 7net-ompa, are compared with DFT results; the slope for 7net-ompa is substantially greater than unity, indicating pronounced PES stiffening. Supplementary Fig. 24 compiles the corresponding distributions for other models and shows that PES stiffening is consistently observed across models, albeit to varying degrees. These position-dependent shifts in the PES imply that simply replacing molecular energies with their corresponding DFT values cannot eliminate the spurious stiffening. By contrast, in the ALD scenario in Fig. 2e, the molecular energy errors cancel out when taking the difference between chemisorbed and physisorbed states, yielding a less pronounced performance gap between models.

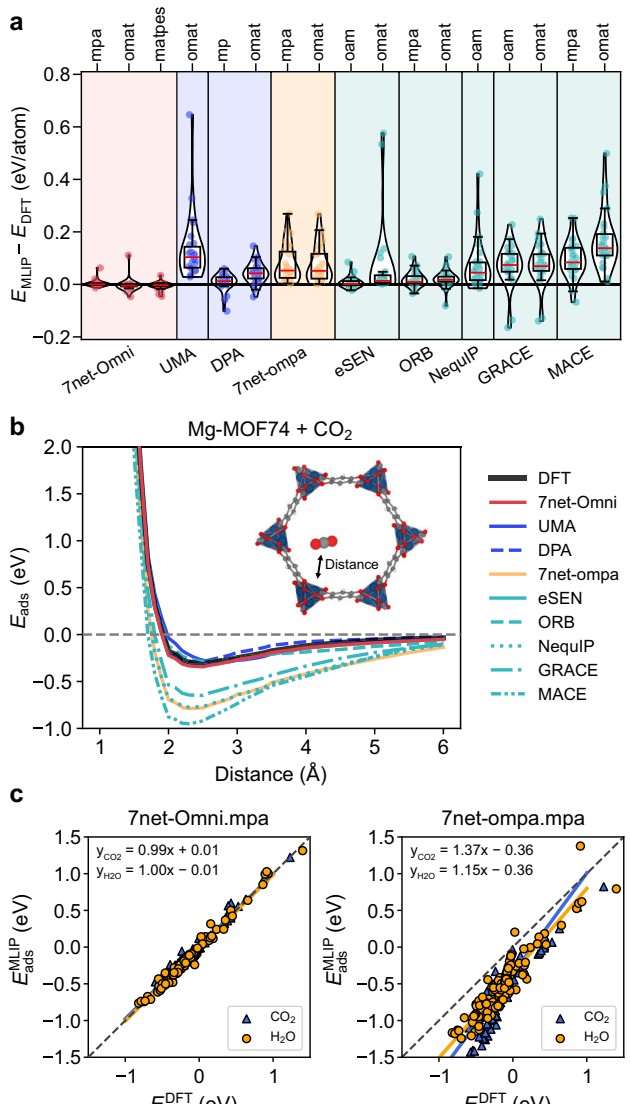

**Fig. 3 | Molecular energy overestimation and PES stiffening. a** Error distributions of molecular energies across uMLIP models, represented as violin and box plots. The inference tasks of multi-task models and the datasets used for single-task models are indicated at the top of the plot. The box and red horizontal line in the box plot show the quartiles and median of the error distribution, respectively, while whiskers represent 1.5 times the inter-quartile range. Individual data points are randomly jittered along the horizontal axis for visual clarity. **b** PES for $CO_2$ adsorption on Mg-MOF74 computed with DFT (PBE-D3) and uMLIPs. **c** Comparison of $E_{ads}$ without deformation for $CO_2$ and $H_2O$ between DFT and 7net models. Linear regression was performed for data points with $E_{ads}^{DFT}$ less than 0.5 eV. The line equations are shown in the figure insets. Source data are provided as a Source Data file.

## Metallic surfaces

Heterogeneous catalysis is another area where computational materials science plays a crucial role. Reactions such as the hydrogen evolution reaction (HER), oxygen evolution reaction (OER), and carbon dioxide reduction reaction (CO$_2$RR) are of major industrial importance, underpinning technologies for solar fuel production and electrocatalytic $CO_2$ conversion[83,84]. Computational studies have provided deep mechanistic insights into these processes[85], while large-scale ab initio datasets, such as those compiled by the Open Catalyst Project[23,24], have enabled the development of catalyst-specific foundation MLIPs. We benchmark our model on a range of metal-surface adsorption tasks, spanning canonical reactions such as HER, OER, and

CO$_2$RR, as well as adsorption of organic molecules on noble-metal surfaces relevant to heterogeneous hydrogenation and oxidation of volatile organic compounds.

We first investigate the adsorption of simple adsorbates (\*H, \*O, \*OH, \*CO) on noble-metal surfaces (Au, Ag, Cu, Pd, Pt). These systems are included in the OC20 dataset, computed with the RPBE functional, and are also sparsely sampled in the DBS (see Supplementary Table 1). For each metal, we consider both the (100) and (111) surfaces, with three symmetric adsorption sites: bridge, hollow, and top for (100) surfaces, and fcc, hcp, and top for (111) surfaces. The adsorption energy of adsorbate X is computed using Eq. (6), where $E^X$ for H, O, and OH is determined from the equilibrium of $H_2$ and $H_2O$ molecules, while for CO the molecular energy is used directly. In total, 120 adsorption energies were computed.

As shown in Fig. 4a, 7net-Omni significantly outperforms other models, achieving a low MAE of approximately 0.06 eV in $E_{ads}$. (See Supplementary Fig. 25 for parity plots). The white bullets on the multi-task models indicate the MAE of the oc20 channel relative to RPBE reference values, showing that the PBE performance of 7net-Omni exceeds that of 7net-Omni.oc20. This may be due to the much larger number of structures available at the PBE level compared with RPBE, which biases the model weights toward PBE fidelity. A similar trend is observed for DPA.omat. By contrast, UMA.omat shows much larger errors than UMA.oc20, indicating inefficient knowledge transfer. (Among various channels of UMA, only UMA.omat was able to produce reasonable results for tests in this section). Single-task models, with the exception of eSEN[oam], fail to achieve comparable accuracy. This is primarily because of their limited ability to describe reference molecular energies, as discussed in the previous section, which leads to systematic shifts in the reaction energies of identical adsorbates. The large errors in eSEN[omat] in comparison with eSEN[oam] are attributed to the inaccurate energies for some molecules, as shown in Fig. 3a.

To extend the evaluation to more diverse metals and adsorbates, we consider the ADS41 dataset[86,87]. ADS41 comprises 15 physisorption systems (primarily organic molecules on noble-metal surfaces) and 26 chemisorption systems. Reference adsorption energies were computed at the PBE-D3 level of theory[87]. The benchmark results are shown in Fig. 4b–d. Figure 4b displays MAEs for physisorption energies, while Fig. 4c presents chemisorption energies excluding Co and Ni. The anomalous Co and Ni systems are displayed separately in Fig. 4d. Compared with Fig. 4a,c includes additional noble metals (Ir, Rh, Ru) as well as adsorbates (CCH$_3$, I, NO). Overall, Fig. 4b,c reveal consistent trends across uMLIPs, with 7net-Omni and eSEN[oam] performing best, in agreement with Fig. 4a. The parity plots for benchmarking ADS41 dataset are provided in Supplementary Fig. 26.

In the previous section, we showed that destabilization of molecular energies significantly affected the PES. To examine such an effect in catalytic reactions, we investigate the CO$_2$RR process on Pt and Pd surfaces. Specifically, we consider COOH formation pathways: $CO_2$ + \*H → \*COOH shown in ref. 88 on Pt(111) and Pd(111) surface, where $CO_2$ is initially physisorbed. Nudged elastic band (NEB) calculations are performed to identify the minimum-energy paths. The resulting PES profiles and error analysis in Supplementary Fig. 27 show that the forward and backward reaction barriers usually differ by 0.1–0.2 eV between DFT and uMLIPs except for 7net-Omni and eSEN[oam], indicating potential inaccuracies in the turn-over frequency of CO$_2$RR reactions.

Figure 4d for Co and Ni shows that most models catastrophically fail to predict $E_{ads}$. This stems from the computational settings used in databases that employ PBE+$U$ (e.g., MPtrj and OMat24). To account for the correlated nature of 3$d$ transition-metal oxides, these databases apply a Hubbard $U$ correction whenever partially filled 3$d$ metals such as Co and Ni coexist with oxygen atoms. (For general conditions for application of PBE+$U$, we refer to ref. 89). As a result, uMLIPs trained

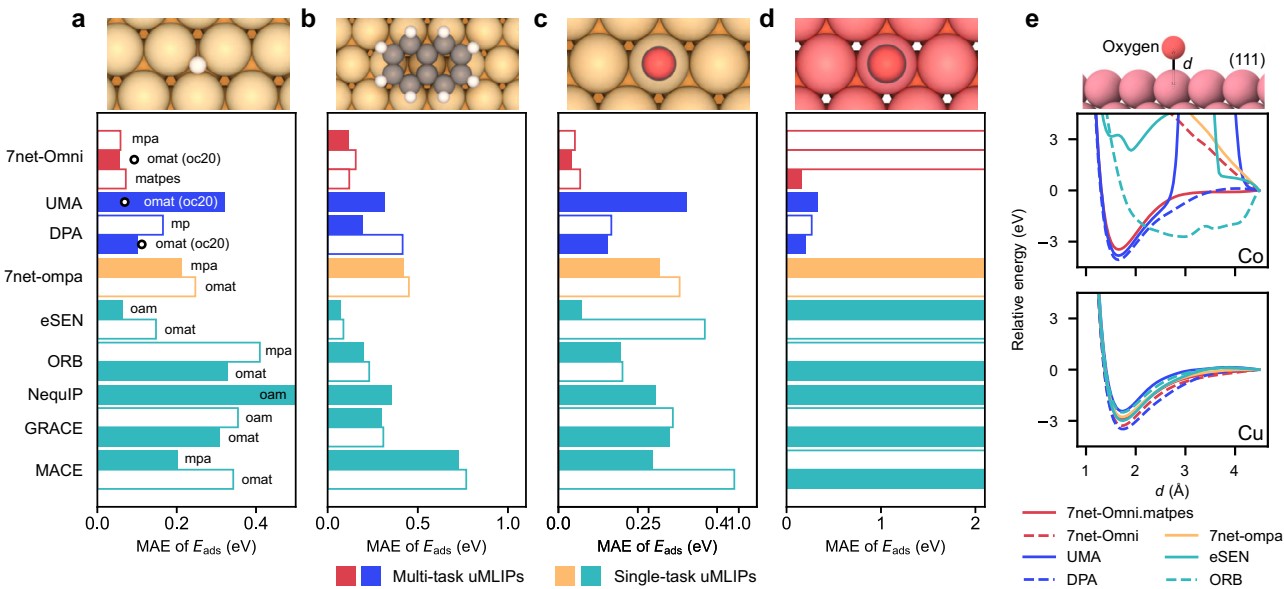

**Fig. 4 | Accuracy of uMLIPs for reactions on metal surfaces. a** MAE for adsorption energies of *H, *O, *OH, and *CO adsorbates on five noble metals, Cu, Pd, Pt, Ag, and Au. The *y*-axis lists the uMLIP models. For multi-task uMLIPs, the inference channel is indicated to the right of each bar; for single-task uMLIPs, the corresponding training set is shown. White bullets mark the performance of the RPBE-fidelity channel (parentheses), compared with RPBE results. **b** MAE of physisorption energy predictions for ADS41 dataset. **c** MAE of chemisorption energy predictions for ADS41 dataset, excluding Co and Ni surfaces. **d** MAE of chemisorption energy predictions for adsorptions on Co and Ni surfaces in ADS41 dataset. **e** Potential energy curves of uMLIPs along the distance between the oxygen atom and the Co or Cu metal surface. Reference DFT data are calculated within PBE (**a**) or PBE-D3 (**b**–**d**). Solid bars in (**a**–**d**) represent the best-performing channel or training database. Individual parity plots are presented in Supplementary Figs. 25, 26. Source data are provided as a Source Data file.

heavily on these databases inherit PESs that include $U$ corrections whenever oxygen atoms interact with Co or Ni.

We illustrate this in Fig. 4e by moving an oxygen atom above Co and Cu(111) surfaces. Most models exhibit anomalous PESs for Co, whereas predictions for Cu remain physically sound. Consequently, the adsorption energies of oxygen-containing molecules on Co and Ni surfaces are widely mispredicted. Notably, the OC20 database includes oxygen-containing adsorbates on partially filled 3$d$ metals but excludes $U$ corrections. However, the weight parameters for mpa and omat channels of 7net-Omni appear more heavily tuned to larger datasets such as MPtrj, sAlex, and OMat24.

Interestingly, this erratic behavior is not observed for 7net-Omni.matpes, which was trained on databases generated without $U$[22] and achieves the best overall performance. In Fig. 4e, 7net-Omni.matpes also produces physically sound PESs. Notably, UMA and DPA maintain robust accuracy despite being trained on MPtrj or OMat24. It remains to be investigated whether the difference arises from the encoding strategy (e.g., DPA encodes task embedding only at the final layer) or from auxiliary mechanisms (e.g., mixtures of linear experts in UMA).

The above discussion is relevant whenever transition metals and oxygen atoms are within the cutoff radius. For example, this occurs in CO binding to single-atom metal centers in organometallic complexes (see Cross-domain or cross-functional scenarios section) and in OH/CHO adsorption on metal nitrides (Supplementary Fig. 28)[90]. These results underscore the need for caution when applying uMLIPs to adsorption on 3$d$ transition-metals.

## r²SCAN fidelity

Recently, the meta-GGA regularized-restored Strongly Constrained and Appropriately Normed (r²SCAN) functional has attracted considerable attention, as it provides improved agreement with experimental results compared to conventional PBE calculations, particularly for properties such as thermal stability, phonons, and lattice volumes[91,92]. However, r²SCAN computations are several times more

expensive than with the PBE functional, and only recently large databases based on r²SCAN become available[22,30,93]. In training 7net-Omni, we included three databases at the r²SCAN level (Table 1). For MP-r²SCAN, data were collected directly from the Materials Project database, while MatPES-r²SCAN and MP-ALOE are directly available in the public domain. The total number of r²SCAN entries (1,283,036) corresponds to only 0.8% of those at the PBE level.

There are two other open models, MACE[22] and VMD[50], that provide r²SCAN energies. MACE is a single-fidelity model trained on MatPES-r²SCAN and MP-ALOE, whereas VMD adopts the multi-fidelity architecture of M3GNet[6,94] to simultaneously train on PBE and r²SCAN data from MatPES. For 7net-Omni, we employ the matpes_r2scan channel instead of mp_r2scan, as the former is trained with larger databases. We also computed the same properties using 7net-Omni.mpa as a baseline. In other words, any performance below 7net-Omni.mpa is regarded as equivalent to having no benefit from training at the r²SCAN level.

The benchmark results for r²SCAN-based uMLIPs are presented in Table 2. We first compute energies of crystals whose SCAN values are provided in the Alexandria database[95]. The crystal structures are fixed and $E_{for}$ with respect to elemental phases is compared. For cross-domain applications involving surface and molecular configurations, we consider the ADS41 and BMCOS1 benchmark sets used in Cross-domain or cross-functional scenarios section, which also provide r²SCAN data. We further evaluate $\kappa$ for binary compounds with high symmetry[42,96] in comparison with theoretical values at the r²SCAN level (see Methods) as well as experimental data[97–99]. Given the growing interest in meta-GGA functionals for predicting ionic conductivity in solid-state electrolytes, we assess model performance on force prediction along the molecular dynamics (MD) trajectories of argyrodite $Li_6PS_5Cl$ reported in ref. 43.

As shown in Table 2, 7net-Omni exhibits the best overall accuracy except for the energy estimation of BMCOS1 dataset. The improved agreement with experimental $\kappa$ for matpes_r2scan channel in comparison with mpa implies that the r²SCAN channel of 7net-Omni can be

**Table 2 | Comparison of uMLIP performance on the r²SCAN fidelity**

| Database | Property | 7net-Omni.matpes_r2scan | MACE | VMD | 7net-Omni.mpa |
|---|---|---|---|---|---|
| Alexandria | $E_{for}$ MAE (eV/atom) | **0.047** | 0.084 | 0.126 | 0.108 |
| ADS41 | $E_{ads}$ MAE (eV) | **0.159** | 0.242 | 0.421 | 0.684 |
| BMCOS1 | $E_{coh}$ (eV/molecule) | 0.190 | 0.270 | - | **0.103** |
| BMCOS1 | Volume MAPE (%) | **0.797** | 4.424 | - | 1.093 |
| Binary solids | $\kappa_{SRE}$ | **0.285** | 0.368 | 0.891 | 0.365 |
| Binary solids (Exp.) | $\kappa_{SRE}$ | **0.242** | 0.334 | 0.877 | 0.348 |
| Li₆PS₅Cl | Force MAE (eV/Å) | **0.024** | 0.078 | 0.205 | 0.065 |

Benchmark databases and accuracy metrics for each predicted physical property are summarized in each row. Three uMLIPs with r²SCAN fidelity are evaluated against the reference r²SCAN data, while the performance of 7net-Omni.mpa is also included as a baseline. The VMD model fails to relax molecular crystals in BMCOS1. $\kappa_{SRE}$ is the mean symmetric relative error for lattice thermal conductivity. The bold number in each row indicates the result of the best performing uMLIP compared to the r²SCAN functional. Individual parity plots are presented in Supplementary Figs. 29–33. Source data are provided as a Source Data file.

utilized to obtain thermal properties of materials more accurately than with the conventional PBE functional. Considering the high computational cost of r²SCAN calculations, this represents a significant advance in theoretical approaches. Parity plots in benchmarking r²SCAN can be found in Supplementary Figs. 29–33.

In Table 2, 7net-Omni.mpa yields more accurate $E_{coh}$ values for molecular crystals than other r²SCAN models. This may be due to the complete absence of molecular structures in the r²SCAN training data. Adding appropriate DBS at the r²SCAN level may improve these results. Likewise, the MAE for $E_{ads}$ is 0.159 eV on the ADS41 dataset, whereas the corresponding MAE in the PBE channel is less than 0.1 eV (see Fig. 4c). Additional DBS in OC20 and OC22 at the r²SCAN level may further improve accuracy.

We also note in Table 2 that 7net-Omni.matpes_r2scan performs substantially better than MACE, despite both models being trained on largely the same r²SCAN databases. The additional MP-r²SCAN set for 7net-Omni contributes only near-equilibrium configurations with limited data size (see Table 1), so the performance gain can be attributed to effective transfer learning from the abundant PBE-level data with similar PES characteristics. This is also consistent with the data-efficient multi-fidelity framework[43].

## Inference speed
We compare the inference speed of various uMLIPs on an NVIDIA H100 GPU (94 GB) using MD simulations of diamond Si structures. The performance results are summarized in Fig. 5, where the *x*-axis denotes the number of atoms and the *y*-axis represents the simulation throughput in nanoseconds per day when the time step is 1 femtosecond. For libraries accelerating tensor-product operations, 7net-Omni was tested with FlashTP[100] (Omni-flash) as well as cuEquivariance (v0.7.0)[101] (Omni-cueq), MACE with cuEquivariance (v0.7.0), and NequIP with OpenEquivariance (v0.4.1)[102]. For models supporting LAMMPS[103] execution, the measurements were performed within LAMMPS, whereas those without LAMMPS support (eSEN, UMA, ORB) were evaluated using ASE MD[104]. Although DPA supports LAMMPS, only the ASE results are reported here due to installation difficulties.

The original SevenNet interface integrated with LAMMPS was used for the FlashTP library[10], while coupling with the cuEquivariance library was achieved through the ML-IAP package within LAMMPS[105]. The lowest precision settings were adopted for ORB (float32-high), UMA (tf32), MACE (float32), and NequIP (float32). For the UMA model, additional acceleration was achieved by applying torch.compile and pre-merging linear experts, while activation checkpointing was enabled for systems containing more than 1000 atoms to fit the model into GPU memory[8]. For the ORB model, torch.compile and cuML-accelerated graph construction[106] were applied for further acceleration. All models were tested with increasing system sizes until the GPU memory limit was reached.

In Fig. 5, it is observed that 7net-Omni, when accelerated with cuEquivariance, achieves a competitive throughput of approximately

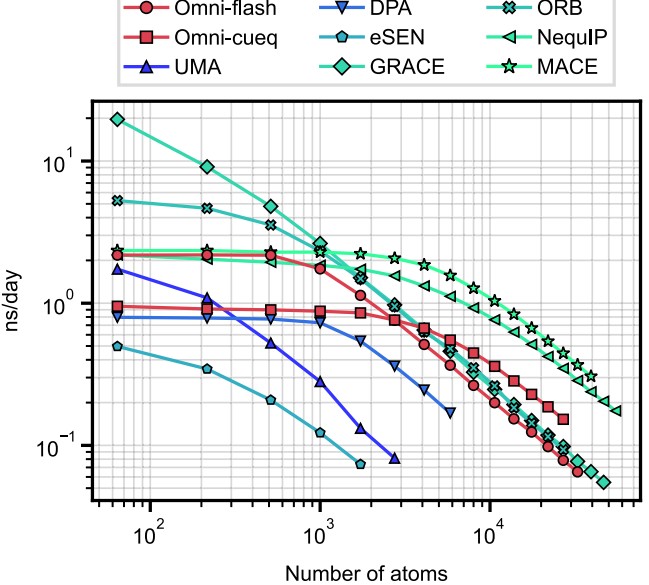

**Fig. 5 | Inference speed of uMLIPs.** The inference performance of each uMLIP is evaluated using MD simulations of diamond Si. The speed is reported in units of nanoseconds per day (ns/day), measured on a H100 GPU card. Source data are provided as a Source Data file.

0.36 ns/day for a system containing 10,000 Si atoms, although this is roughly one-third of the speed of MACE, the fastest model in the present benchmark. In contrast, eSEN, which attains accuracy comparable to that of 7net-Omni in many cases, requires significantly more memory and demonstrates substantially slower performance. While cuEquivariance provides superior inference performance for 7net-Omni compared to FlashTP acceleration in larger systems (> 3000 atoms), its throughput decreases for smaller systems. This decline is likely due to GPU underutilization or additional auxiliary overhead, indicating potential for further optimization. Notably, GRACE maintains high efficiency even for small system sizes.

## Methods
### Training set
For training 7net-Omni, we employ the databases listed in Table 1. Specifically, we use the full datasets from MPtrj, OMat24, MatPES, OC22, MAD, MatPES-r²SCAN, and MP-ALOE, while subsampling data from Alex, OC20, ODAC23, OMol25, SPICE, QCML, and MP-r²SCAN.

The Alex database consists of relaxation trajectories of 3D, 2D, and 1D structures. For 3D structures, we adopt the sAlex split provided in ref. 21. For 2D and 1D structures, we follow the subsampling procedure in ref. 21, excluding outliers with total energy above 0 eV,

atomic forces exceeding 50 eV/Å, or stress greater than 80 GPa. From the remaining trajectories, we sample relaxation snapshots with energy differences greater than 10 meV/atom.

The OC20 database contains relaxation trajectories, ab initio MD configurations, and randomly displaced (rattled) structures. For relaxation trajectories, we use the OC20M split described in ref. 23. For MD and rattled structures, we exclude configurations with total energy above 0 eV, atomic forces larger than 20 eV/Å, or isolated atoms without any neighbor within 6 Å; a typical cutoff radius in many uMLIPs. We further subsample MD trajectories by selecting every fifth timestep and randomly selecting 20% of rattled structures.

From the ODAC23 database, we subsample relaxation trajectories by selecting snapshots with energy differences greater than 10 meV per adsorbate. For the MP-r²SCAN database, we utilize the final snapshots of all relaxation trajectories provided in the Materials Project[93,107] (retrieved in April 2025).

The OMol25, SPICE, and QCML databases include molecular configurations generated with varying charge and spin multiplicities. We first exclude all structures with a nonzero total charge. For the SPICE dataset, we filtered out configurations with a maximum atomic force exceeding 15 eV/Å or a total force exceeding 0.1 eV/Å, following the same screening criteria as in ref. 33. Both SPICE and QCML contain only the lowest possible spin configurations for each system (i.e., singlets for even-electron systems and doublets for odd-electron systems), and thus we retain all charge-neutral structures from these databases. In contrast, OMol25 also provides ab initio data for highest spin configurations, primarily to account for transition-metal complexes. Accordingly, we divide OMol25 into two subsets corresponding to the lowest and highest spin states.

Notably, SPICE is recalculated as part of OMol25, leading to duplicate structures between the two datasets. Nevertheless, we treat SPICE as a separate training task because it provides DFT energies and forces without dispersion corrections. This distinction is useful when dispersion corrections are added to uMLIPs, particularly for low-density systems with long-range interactions beyond the cutoff radius of uMLIPs.

For DBS generation for training 7net-Omni, we subsample structures from databases that are both representative of each material domain and sufficiently extensive, namely MatPES, OC20, OC22, ODAC23, OMol25, and QCML, totaling approximately 125 million structures. From these, we select about 125,000 structures (0.1%) to construct the DBS.

For sampling, since training 7net-Omni starts with the weights of 7net-ompa (see the next section), we do not uniformly sample 0.1% from each database. Instead, we assign higher sampling weights to structures that are (i) structurally distant from inorganic systems and (ii) derived from databases whose computational settings differ significantly from the PBE functional used in OMat24 and MPtrj. This is quantified by the force prediction error of 7net-ompa on each database, since the resulting errors reflect contributions from both (a) structural dissimilarity with respect to the 7net-ompa training set and (b) differences between the ab initio methods used to label the data and PBE. The resulting sampling ratios are summarized in Supplementary Table 1. After assigning total sampling numbers for each database, we randomly subsample the structures from each database. For ab initio calculations, we employ the Vienna Ab initio Simulation Package (VASP)[108] together with the MPRelaxSet from the pymatgen package to ensure consistency with the computational setups of MPtrj and Alex. We do not apply Hubbard $U$ corrections for structures sourced from OC20, ODAC23, OMol25, and QCML.

To monitor and evaluate the generalizability toward real-world applications of the trained model, a subset of data points is held out from the training process. Specifically, we use the original train/validation split when available; otherwise, we randomly sample 5% of the data from the original database and exclude them from training.

## Training

For training 7net-Omni, we employ a curriculum learning strategy that progressively expands the training set from inorganic bulk systems to organic molecules. We begin by training 7net-MF on MPtrj, sAlex, and OMat24, resulting in the intermediate model 7net-ompa. Next, we extend the training set by incorporating OMol25, which provides the most extensive molecular data, together with the DBS structures derived from OMol25. Finally, we include all databases listed in Table 1 and train 7net-Omni for a single epoch to complete the unified model. When training 7net-Omni, we apply oversampling to certain databases to achieve a balanced representation across material domains. The oversampling ratios are chosen such that the effective number of training structures in inorganic crystals, catalytic surfaces, and molecular systems becomes comparable. For each material class, the ratio is determined by considering both the number of structures and the average number of atoms per structure.

For the model architecture, we set the maximum order of spherical harmonics ($l_{max}$) to 3 for all equivariant node features, employing five convolution layers. The dimensionalities of node features are 128, 64, and 32 for $l = 0$, $l = 1$, and $l > 1$ components, respectively. The model distinguishes between even and odd parity in node features and incorporates all possible combinations of $l$ and parity in constructing tensor products.

For training, we employ the Adam optimizer with an initial learning rate warm-up to a maximum value of 0.002, followed by a smooth decay to zero within one epoch using a cosine annealing schedule. The training loss is defined as follows:

$$
\begin{aligned}
\mathcal{L} = &\frac{\lambda_E}{M} \sum_{i=1}^{M} \frac{|\widehat{E}_i - E_i|}{N_i} \\
&+ \frac{\lambda_F}{M \sum_i^M N_i} \sum_{i=1}^{M} \sum_{j=1}^{N_i} \sqrt{\sum_{k=1}^{3} |\widehat{F}_{i,j,k} - F_{i,j,k}|^2} \\
&+ \frac{\lambda_S}{M} \sum_{i=1}^{M} \sqrt{\sum_{j=1}^{3} \sum_{k=1}^{3} |\widehat{S}_{i,j,k} - S_{i,j,k}|^2} \\
&+ \frac{\lambda_R}{2} \sum |\boldsymbol{\theta}_T|^2,
\end{aligned}
\tag{7}
$$

where the loss weights $\lambda_E$, $\lambda_F$, $\lambda_S$, and $\lambda_R$ correspond to the energy, force, stress, and regularization terms, respectively. Here, $M$ denotes the number of structures in a batch, and $N_i$ represents the number of atoms in the $i$-th structure. The quantities $E_i$, $F_{i,j,k}$, and $S_{i,j,k}$ refer to the ab initio energy, the $k$-th Cartesian component of the force on the $j$-th atom, and the $(j, k)$ component of the symmetric Cauchy stress tensor of the $i$-th structure, respectively. The corresponding predictions from the MLIP are denoted as $\widehat{E}_i$, $\widehat{F}_{i,j,k}$, and $\widehat{S}_{i,j,k}$. The weighting factors are set to $\lambda_E = \lambda_F = 1$, $\lambda_S = 10^{-4}$, and $\lambda_R = 10^{-6}$, with energy, force, and stress expressed in units of eV, eV/Å, and kbar, respectively.

The choice of loss weight ratios plays a critical role in determining model performance. For the energy, force, and stress loss weights, we adapted the values used in ref. 12 to account for differences in both units and loss definitions. We further observe that the regularization loss weight (denoted as $\lambda_R$ in Eq. (7)) has a pronounced impact on MLIP performance, particularly in cross-domain applications. To identify an appropriate value of $\lambda_R$, we trained SevenNet-MF models on the MPtrj and SPICE datasets using a range of $\lambda_R$ values. The resulting performance is summarized in Supplementary Fig. 34. Supplementary Fig. 34a presents the validation errors together with the L2 norm of the task-specific parameters, in analogy to Fig. 1d. The case $\lambda_R = 0$ corresponds to training without regularization. As $\lambda_R$ increases, the norm of $\boldsymbol{\theta}_T$ decreases as expected, leading to improved performance on Molecule@GGA examples when $\lambda_R$ is in the range of approximately

$10^{-5}$–$10^{-4}$. In contrast, using an excessively large regularization strength ($\lambda_R = 10^{-3}$) results in a rapid degradation of model accuracy.

The origin of these errors was further investigated by analyzing the potential energy surface associated with water dimer binding. As shown in Supplementary Fig. 34b, when the regularization strength is too small ($\lambda_R = 10^{-7}$), SevenNet-MF fails to capture a meaningful common PES, similar to the behavior observed for the unregularized model in Supplementary Fig. 2a. In contrast, an excessively large regularization strength ($\lambda_R = 10^{-3}$) overly suppresses task-specific contributions, preventing the model from learning physically meaningful differences between levels of theory. Based on these observations, we tested $\lambda_R$ values in the vicinity of $10^{-5}$ when training 7net-Omni and selected the optimal value accordingly. We note that the optimal choice of $\lambda_R$ may depend on the specific model architecture and training dataset, and that similar hyperparameter tuning, as illustrated in Supplementary Fig. 34, is likely necessary in general.

Another key requirement for multi-task training is the alignment of total energies across databases, which differ due to variations in atomic reference energies. Ref. 40 employed atomization energies to align the total energies of the MPtrj and SPICE databases for concurrent mixed-fidelity training of the MACE-Osaka24 model. Similarly, ref. 8 utilized atomization energies and heats of formation to train the multi-fidelity UMA model across heterogeneous datasets. In contrast, ref. 109 adopted a purely statistical approach, fitting atomic reference energies based on elemental composition. This method, applicable even when energies of isolated atom are unavailable for specific ab initio methods, demonstrated improved transferability between GGA and r²SCAN functionals.

Following these studies, the handling of differing atomic energy references across databases was also implemented in 7net-MF, which supports task-wise energy shifts and scaling. Following the observation in ref. 43, we apply only task-wise shifts while maintaining a universal scaling factor across all tasks. In this work, we adopt a modified version of the statistical approach from ref. 109, as calculating isolated-atom energies for all methods represented in Table 1 would be computationally cumbersome. During curriculum learning, the shift parameters ($s$) for newly added databases are initialized via linear regression using Eq. (8)

$$s = (N^\top N)^{-1} N^\top (E - \sigma \widehat{E}) \tag{8}$$

with

$$E = \begin{pmatrix} E_{DFT}(\mathcal{G}_1) \\ E_{DFT}(\mathcal{G}_2) \\ \vdots \\ E_{DFT}(\mathcal{G}_m) \end{pmatrix}, \widehat{E} = \begin{pmatrix} \widehat{f}(\mathcal{G}_1; \boldsymbol{\theta}_C, \mathbf{0}) \\ \widehat{f}(\mathcal{G}_2; \boldsymbol{\theta}_C, \mathbf{0}) \\ \vdots \\ \widehat{f}(\mathcal{G}_m; \boldsymbol{\theta}_C, \mathbf{0}) \end{pmatrix}, \tag{9}$$

and

$$N = \begin{pmatrix} n_{1,1} & n_{1,2} & \cdots & n_{1,a} \\ n_{2,1} & n_{2,2} & \cdots & n_{2,a} \\ \vdots & \vdots & \vdots & \vdots \\ n_{m,1} & n_{m,2} & \cdots & n_{m,a} \end{pmatrix}, \tag{10}$$

where $n_{ij}$ denotes the number of atoms of the $j$-th element in the $i$-th structure, $a$ and $m$ represent the total number of element types used in the model and structures in the dataset, respectively, and $\sigma$ is the universal scaling factor. $E_{DFT}(\mathcal{G}_i)$ indicates the DFT reference energy of the $i$-th atomic configuration $\mathcal{G}_i$, while $\widehat{f}(\mathcal{G}_i; \boldsymbol{\theta}_C, \mathbf{0})$ represents the value of the common PES predicted by the MLIP prior to applying the shift-scale correction. The shift parameters are kept trainable throughout the subsequent stages of training.

To achieve faster and more memory-efficient inference and training, the convolution layers of SevenNet utilized the FlashTP CUDA kernel, yielding about a fourfold increase in throughput compared to the e3nn library[100,110].

## DFT calculation on benchmark datasets

For single-domain applications, all reference DFT data are taken directly from the original studies, and detailed computational settings can be found in those sources (see citations in Single-domain applications section).

The ADS41 benchmark dataset computed at the PBE-D3 and r²SCAN levels of theory is obtained from ref. 87. Adsorption reaction data for ASD inhibitors are provided by the authors of ref. 76. The BMCOS1 molecular crystal data, including results at both PBE-D3 and r²SCAN-D3 levels, are obtained from ref. 71. For the X23 dataset, optimized cell volumes and cohesive energies at the PBE-D3 level are collected from ref. 70 and combined with BMCOS1 to construct the molecular crystal benchmark.

For additional in-house calculations in the below, VASP[108] is used with ENCUT of 520 eV. When required, Grimme's D3 dispersion correction with Becke-Johnson (BJ) damping is applied[111,112].

For the noble-metal adsorption benchmark in Fig. 3a, metal slabs with five atomic layers are modeled. Lattice constants are determined from bulk crystal relaxations using an atomic force threshold of 0.01 eV/Å. The same force convergence criterion is applied for the optimization of pristine slab, adsorbate-slab, and reference molecules. The bottom two layers are fixed during slab and adslab relaxations. A similar procedure is applied to the NEB calculations for the $CO_2$RR process on Pt(111) and Pd(111) surfaces. Specifically, the initial and final states are relaxed with a force threshold of 0.02 eV/Å, followed by a climbing-image NEB calculation using seven intermediate images and a force convergence criterion of 0.05 eV/Å.

Dihedral-angle-constrained ionic relaxations are performed using VASP interfaced through the Atomic Simulation Environment (ASE) package[104] for the molecular torsion benchmark. A vacuum spacing of 10 Å and dipole corrections along all three Cartesian axes are applied to eliminate interactions between periodic images. Geometry optimizations are conducted until all atomic forces are below 0.01 eV/Å. The same vacuum padding and dipole corrections are applied in the optimization of organometallic complexes, which are converged within a force threshold of 0.02 eV/Å.

For the lattice thermal conductivity benchmark at the r²SCAN fidelity, we follow the computational protocol and ab initio settings described in refs. 97,98, with two modifications: (i) the XC functional is replaced with r²SCAN, and (ii) the plane-wave cutoff energy is increased from 520 eV to 680 eV, as the r²SCAN functional generally requires a higher cutoff to ensure convergence[113,114] compared to GGA. To generate displaced structures for second- and third-order force constants and to calculate lattice thermal conductivity, we employ the phono3py package[96].

## Details of benchmark calculations

The ASE calculator interface is used for all uMLIP evaluations[104]. For the molecular torsion, organometallic reactions, ASD adsorption, molecular crystal, and ADS41 benchmarks, the DFT-optimized geometries serve as the initial structures for uMLIP relaxations[61,67–69,71,87]. Other evaluations requiring optimizations such as non-crystalline benchmarks, noble metal adsorption, and MOF properties follow the DFT data construction process, which can be found in the original study[55,58,79–82] or the previous sections. For the Henry coefficient calculations of MOFs, 10,000 cycles are iterated using the process provided in ODAC25 study[31]. D3 dispersion corrections with BJ damping are consistently applied where appropriate through the internal implementation in SevenNet, which supports damping parameters for PBE[112], r²SCAN[115], and $\omega$B97M[47] XC functionals. The force convergence

criteria for uMLIP relaxations are set to match those used in the corresponding DFT reference calculations, except for the BMCOS1 benchmark, where the threshold is set to 0.005 eV/Å following X23 study[70], and for the defect binding energy benchmark, where a relaxed threshold of 0.01 eV/Å is applied.

## Data availability

The DBS database, in-house DFT calculation results and some benchmark results that are too extensive to be provided as Source Data file generated in this study have been deposited in the figshare under accession code 30399814 https://doi.org/10.6084/m9.figshare.30399814[116]. The DFT reference and experimental data utilized in this study are available at refs. [31,55–57,59,60,70,71,80,81,86,95,97–99]. Source data are provided with this paper.

## Code availability

The SevenNet source code for training and evaluation is available at Zenodo (https://doi.org/10.5281/zenodo.18505985)[117] and Github (https://github.com/MDIL-SNU/SevenNet).

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

## Acknowledgements

We thank Euitae Lee and Hyuntae Cho for their support in the ML-IAP integration of SevenNet into LAMMPS, the NVIDIA cuEquivariance team for their support in the cuEquivariance implementation, and Gijin Kim for discussion on some benchmarks. This work was supported by the Neural Processing Research Center program of Samsung Electronics Co., Ltd. (S.H.), and the Virtual Engineering Platform Project of the Ministry of Trade, Industry and Energy (MOTIE) of Korea (grant number: P0022336; S.H.). This research was supported by the High-Performance Computing Support Project, funded by the Government of the Republic of Korea (Ministry of Science and ICT, RQT-25-070256; S.H.). The computations were carried out at the Center for Advanced Computations (CAC) at Korea Institute for Advanced Study (KIAS), Korea Institute of Science and Technology Information (KISTI), National Supercomputing Center (KSC-2025-CRE-0284; S.H.).

## Author contributions

J.K.[1], J.Y., and S.H. devised and formalized the idea. J.K.[1] and Y.P. developed the training code base. J.K.[1], J.Y., Y.L., Y.K.[1], J.K.[2], H.J., S.J., and D.H. benchmarked the model performance. S.Y.L., Y.P., and J.W.L. developed and integrated the acceleration of tensor-product operations. J.Y., S.C., and Y.K.[2] contributed to the discussion of inference throughput. J.K.[1], J.Y., Y.L., and S.H. prepared the manuscript. All authors contributed to discussions and approved the paper. (J.K.[1]: Jaesun Kim, J.K.[2]: Jisu Kim, Y.K.[1]: Yujin Kang, Y.K.[2]: Yongdeok Kim).

## Competing interests

The authors declare no competing interests.
