## [Transparent Peer Review file · Nature Communications]

Optimizing Cross-Domain Transfer for Universal Machine Learning Interatomic Potentials

Corresponding Author: Professor Seungwu Han

Version 0:

Reviewer comments:

Reviewer #1

(Remarks to the Author)

The paper is concerned with a new multi-domain training strategy for machine learning interatomic potentials (MLIPs). Here, the main purpose is to optimize the cross-domain (knowledge) transfer. To this end, the authors propose at first hand a new selective task regularization strategy. Moreover, a domain-bridging set is introduced. These approaches are applied to train a multi-domain MLIPs SevenNet-Omni based on the SevenNet architecture and on 15 open datasets spanning molecules, crystals, and surfaces. The authors benchmark this new model versus several state of the art pretrained universal MLIPs with respect to accuracy and performance in various applications.

The paper is very interesting and well written. The presented results appear to be plausible and consistent with the discussions and conclusions drawn in the text. Their results offer a scalable route towards more universal, transferable MLIPs that bridge different fidelities and chemical domains.

Therefore I recommend the paper to be published, with only some minor modifications to address and clarify the following comments:

i. The authors refer in equation (2) the conventional Taylor expansion, with also the remainder terms in (3).

a) At least from a mathematical point of view to apply the Taylor Expansion with its error estimates to be valid, the function has to fulfill some smoothness conditions. This is not mentioned or discussed.

b) It is maybe not completely clear to the reader and hence maybe confusion for the reader, if the is just a motivation, or if the additive expansion (20 is somehow used in the architecture of the model. Moreover, e.g. the equation (3) is as far as I could see never used again in the paper. Hence, the question is if this expansion is just to motivate or is it the basis to rigorously drive the selective task regularization approach. Or maybe it could be used for it's analysis from more mathematical point of view.

ii. the authors may clarify in more detail how they have chosen the precise value of the regularization parameters and also of the other involved loss weights.

iii. Figure 3 a): there is a blue area with UMA and DPA, but there is σ_{mat} , μ_p and σ_{mat} , which is somehow confusing ... (maybe there is a separating line, but at least I could not see one in my printed version)

(Remarks on code availability)

Reviewer #2

(Remarks to the Author)

Review of "Optimizing Cross-Domain Transfer for Universal Machine Learning Interatomic Potentials"

Summary This manuscript presents SevenNet-Omni, a universal machine learning interatomic potential (uMLIP) trained on a massive and diverse aggregation of 15 datasets. The authors propose a novel training strategy utilizing multi-task learning with selective regularization of task-specific parameters and the inclusion of a Domain-Bridging Set (DBS) to align potential energy surfaces across heterogeneous computational protocols.

I find the core contributions—specifically the regularization strategy and the DBS—to be both novel and highly useful for addressing the critical problem of cross-domain generalization. The benchmarking campaign is particularly impressive and valuable due to its sheer scale, covering a vast array of tasks across inorganic, organic, and interfacial systems. Notably, the model's ability to outperform specialized models—such as surpassing dedicated r2SCAN-trained models on r2SCAN benchmarks—is a particularly promising result that validates the proposed transfer learning approach.

I believe this work is of high quality and worth publishing; however, there are several comments regarding the analysis and specific benchmarking baselines that should be addressed first:

Comments

1) Analysis of Regularization and Task-Specific Parameters: The authors provide an interesting analysis based on a Taylor expansion around the minimum where all task-specific parameters (θ_T) are set to 0. The theoretical implication is that regularization forces the "common" parameters (θ_C) to learn a more generalizable representation. It would be very valuable to see empirical validation of this: How does the model actually perform if you set all task-specific parameters to 0 during inference? Is it an accurate general model as the analysis implies? This would strongly support the claim regarding the shared representation.

2) Curriculum Learning Ablation: The training procedure follows a specific curriculum: Crystal \rightarrow Crystal+Molecule \rightarrow All. It is unclear if this specific order is necessary or if it introduces bias. Did the authors attempt joint training on all datasets from scratch? An ablation study (or at least a comment) on whether the curriculum order prevents "catastrophic forgetting" of the initial crystal stability would be valuable.

3) Contradiction in DBS Sampling Strategy: There appears to be a contradiction in how the Domain-Bridging Set (DBS) construction is described. In the main text (Page 4), the authors state they "randomly subsample" structures. However, in the Methods section (Section 3.1), they state they "assign higher sampling weights to structures that are (i) structurally distant from inorganic systems...". This distinction is critical: "random" sampling implies the method works without active selection, whereas "weighted" sampling implies an active learning or adversarial approach is necessary for success. The authors must clarify which method was actually used and, if weighted, how the "structural distance" was quantified.

4) Clarification of "Performance Improvement": The manuscript frequently uses phrases such as "yields a substantial performance improvement." I recommend the authors be careful with this wording to clearly distinguish between accuracy with respect to the training labels (PBE/DFT) and accuracy with respect to physical reality. Improved performance here implies higher fidelity to the PBE level of theory, but due to the limitations of GGA functionals, this does not necessarily translate to improved accuracy in reality

5) ORB Benchmarking and Speed: In Figure 5, the inference speed numbers reported for ORB are surprisingly low and appear to disagree with the results reported in the ORB-v3 paper. This discrepancy suggests that the ASE implementation used here may not have been properly optimized. I strongly suggest contacting the ORB developers to help debug this or verify the setup to ensure a fair comparison.

6) Claims regarding ORB Equivariance: The text states: "ORB employs equivariant vector features but does not strictly enforce rotational equivariance throughout the model." I believe this statement may be incorrect. I request the authors double-check the technical details of the specific ORB version used or clarify exactly what is meant by "does not strictly enforce," as this could be misleading.

7) Figure 1c: The blue solid line should be explicitly defined in the caption

8) Figure 1b: It is not clear what the "task embedding vectors" are or how they were derived.

9) Figure 3c: Please clarify exactly what data set is being used here

10) Figure 3a: The performance of UMA on OMat seems surprisingly poor. I would double-check this result or clarify if there is a specific reason for this underperformance.

(Remarks on code availability)

Version 1:

Reviewer comments:

Reviewer #1

(Remarks to the Author)

I have reviewed the revised manuscript and the authors' responses to my comments. I am satisfied with all the changes

made and recommend this manuscript for acceptance.

(Remarks on code availability)

Reviewer #2

(Remarks to the Author)

I am satisfied with the changes made in response to the reviewer comments

(Remarks on code availability)

Reviewer #1 (Remarks to the Author):

The paper is concerned with a new multi-domain training strategy for machine learning interatomic potentials (MLIPs). Here, the main purpose is to optimize the cross-domain (knowledge) transfer. To this end, the authors propose at first hand a new selective task regularization strategy. Moreover, a domain-bridging set is introduced. These approaches are applied to train a multi-domain MLIPs SevenNet-Omni based on the SevenNet architecture and on 15 open datasets spanning molecules, crystals, and surfaces. The authors benchmark this new model versus several state of the art pretrained universal MLIPs with respect to accuracy and performance in various applications.

The paper is very interesting and well written. The presented results appear to be plausible and consistent with the discussions and conclusions drawn in the text. Their results offer a scalable route towards more universal, transferable MLIPs that bridge different fidelities and chemical domains.

Therefore I recommend the paper to be published, with only some minor modifications to address and clarify the following comments:

I. The authors refer in equation (2) the conventional Taylor expansion, with also the remainder terms in (3).

a) At least from a mathematical point of view to apply the Taylor Expansion with its error estimates to be valid, the function has to fulfill some smoothness conditions. This is not mentioned or discussed.

b) It is maybe not completely clear to the reader and hence maybe confusion for the reader, if the is just a motivation, or if the additive expansion (2) is somehow used in the architecture of the model. Moreover, e.g. the equation (3) is as far as I could see never used again in the paper. Hence, the question is if this expansion is just to motivate or is it the basis to rigorously drive the selective task regularization approach. Or maybe it could be used for it's analysis from more mathematical point of view.

Response: We appreciate the reviewer for pointing this out. a) To apply equation (2) for function $f(\mathcal{G}; \theta_C, \theta_T)$, f should be smooth (i.e. C^∞) for θ_T . This is mostly true for physical MLIPs that ensure smooth PES with \mathcal{G} , thus MLIPs usually employ smooth activation functions which also makes f become smooth function for θ_T . Activation functions in MLIP models should be at least C^1 functions to ensure the continuity of force, and popular activation functions such as ReLU are avoided in the field of MLIP for this reason. (doi.org/10.1021/acs.chemrev.0c01111) Here, we revise our discussion to MLIPs that infer continuous force, which indicates f would be at least C^1 function for θ_T . By applying Taylor's theorem for C^1 function, we can write that

$$f(\mathcal{G}; \theta_C, \theta_T) = f(\mathcal{G}; \theta_C, 0) + \theta_T^\top \cdot \nabla_{\theta_T} f(\mathcal{G}; \theta_C, 0) + \theta_T^\top \cdot R_1(\mathcal{G}; \theta_C, \theta_T) = f(\mathcal{G}; \theta_C, 0) + \theta_T^\top \cdot R(\mathcal{G}; \theta_C, \theta_T)$$

and

$$R(\mathcal{G}; \theta_C, \theta_T) := \int_0^1 \nabla_{\theta_T} f(\mathcal{G}; \theta_C, t \theta_T) dt$$

since the derivative of f by θ_T is continuous. This shows that the result of the equations in our original manuscript holds for typical MLIP architecture which ensures the continuity of inferred force.

b) Throughout the discussion, terms in (2) and (3) are not explicitly utilized in the architecture of the model. It is rather a motivation for employing selective regularization. Since the derivative of f by θ_T is continuous, the absolute value of equation (3) has an upper bound, which indicates the task-specific contribution term in equation (2) can be suppressed by regularizing the size of θ_T , leading the model to rely more on common PES. We agree with the reviewer that discussion of the smoothness and further explanation of our motivation on the regularization strategy would be helpful for the readers' understanding.

Actions taken:

On page 3, we have added two consecutive sentences starting with "For the continuity of inferred force ...".

On page 3, we have revised the sentence starting with "Therefore, one can apply Taylor's theorem to ...".

On page 3, we have revised the Eq. 2.

On page 4, we have added the sentence starting with "Motivated by Eq. 2, the task-specific contribution ...".

II. The authors may clarify in more detail how they have chosen the precise value of the regularization parameters and also of the other involved loss weights.

Response: We thank the reviewer for raising this point. The loss weights for energy, force, and stress were chosen by referencing values commonly used in training other universal MLIPs, with minor adjustments to account for differences in units and loss definitions. For example, eSEN-MPTrj employs an energy:force:stress loss weight ratio of 1:1:0.25, where the corresponding units are eV/atom, eV/Å, and eV/Å³, and the losses are defined using MAE, L2MAE, and MAE, respectively. (doi.org/10.48550/arXiv.2502.12147v2) Since SevenNet uses the same units and loss definitions for energy and force, we adopted the same relative weighting of 1:1 for these two terms. In contrast, SevenNet treats stress using L2MAE with units of kbar, which typically yields stress loss values several orders of magnitude larger than those obtained using MAE in eV/Å³. To compensate for this difference in scale, we reduced the stress loss weight from the reference value of 0.25 to 1×10^{-4} .

After choosing the loss weights for energy, force, and stress, we investigated the effect of different regularization strengths (λ_R) when training multi-task SevenNet. We first identified a suitable range for λ_R by training models on the MPtrj and SPICE datasets. The force MAEs on the validation set used in Fig. 1d of the original manuscript, together with the resulting PES for the water dimer example, are shown in Fig. R1.

Fig. R1. Performance of multi-task models by varying regularization loss weight (λ_R). **a** Force MAE and L2 norm trend with selection of λ_R . Blue and purple plots show force MAE in MPtrj channel compared with the PBE reference for crystal and molecular structures, respectively. The red plot shows the force MAE in the SPICE channel compared with the ω B97M reference. The yellow plot shows the L2 norm of task-specific parameters decreases as λ_R increases. **b** Potential energy surface of water dimer binding. Blue and red lines show PES calculated with MPtrj channel and SPICE channel, respectively. Gray dashed lines show common PES contributions, obtained with inferring models by setting task-specific parameters to zeros. The blue shaded region shows the task-specific contribution for MPtrj channel. Blue and red markers indicate reference binding energy calculated with corresponding *ab initio* methods.

As shown in Fig. R1a, the error for Molecule@GGA increases when λ_R is either too large (1×10^{-3}) or too small ($< 1 \times 10^{-7}$). The PES in Fig. R1b further indicates that overly large λ_R excessively suppresses task-specific contributions, preventing the model from capturing meaningful differences between levels of theory, while overly small λ_R fails to recover an appropriate common PES, as indicated by the gray dashed curves. Consequently, large discrepancies between the reference PBE energies (blue markers) and the predicted PES (blue lines) are observed for $\lambda_R = 1 \times 10^{-3}$ and 1×10^{-7} , whereas an intermediate value of $\lambda_R = 1 \times 10^{-5}$ yields reasonable agreement. Based on these observations, we tested $\lambda_R \in \{1 \times 10^{-4}, 1 \times 10^{-5}, 1 \times 10^{-6}\}$ when training SevenNet-Omni and found $\lambda_R = 1 \times 10^{-6}$ to be optimal for our final architecture and datasets. Notably, the optimal value for λ_R may vary by the choice of model architecture and the distribution of training set while λ_R near 1×10^{-5} is in general, a good starting point.

Actions taken:

On page 16, we have added two consecutive paragraphs starting with “The choice of loss weight ratios plays a critical role in ...”.

In Supplementary Information, we have added the Fig. R1 as Supplementary Fig. 32.

III. Figure 3 a): there is a blue area with UMA and DPA, but there is omat, mp and omat, which is somehow confusing ... (maybe there is a separating line, but at least I could not see one in my printed version)

Response: We have revised Figure 3a by making the separating lines between the models more distinct to improve the visual clarity.

Actions taken:

On page 11, we have revised Fig. 3a.

Reviewer #2 (Remarks to the Author):

Review of "Optimizing Cross-Domain Transfer for Universal Machine Learning Interatomic Potentials"

Summary

This manuscript presents SevenNet-Omni, a universal machine learning interatomic potential (uMLIP) trained on a massive and diverse aggregation of 15 datasets. The authors propose a novel training strategy utilizing multi-task learning with selective regularization of task-specific parameters and the inclusion of a Domain-Bridging Set (DBS) to align potential energy surfaces across heterogeneous computational protocols.

I find the core contributions—specifically the regularization strategy and the DBS—to be both novel and highly useful for addressing the critical problem of cross-domain generalization. The benchmarking campaign is particularly impressive and valuable due to its sheer scale, covering a vast array of tasks across inorganic, organic, and interfacial systems. Notably, the model's ability to outperform specialized models—such as surpassing dedicated r2SCAN-trained models on r2SCAN benchmarks—is a particularly promising result that validates the proposed transfer learning approach.

I believe this work is of high quality and worth publishing; however, there are several comments regarding the analysis and specific benchmarking baselines that should be addressed first:

Comments

1) Analysis of Regularization and Task-Specific Parameters: The authors provide an interesting analysis based on a Taylor expansion around the minimum where all task-specific parameters (θ_T) are set to 0. The theoretical implication is that regularization forces the "common" parameters (θ_C) to learn a more generalizable representation. It would be very valuable to see empirical validation of this: How does the model actually perform if you set all task-specific parameters to 0 during inference? Is it an accurate general model as the analysis implies? This would strongly support the claim regarding the shared representation.

Response: We thank the reviewer for the insightful suggestion. In response, we evaluated several multi-task SevenNet models trained on the MPtrj and SPICE databases (the same models used in Fig. 1d of the original manuscript) on a potential energy surface (PES) associated with the binding of two water molecules. The results are summarized in Fig. R2. In this figure, the blue and red solid lines represent the inferred PES of the multi-task SevenNet models with task-specific parameters θ_T trained on MPtrj and SPICE, respectively (hereafter referred to as SevenNet.MPtrj and SevenNet.SPICE). The gray dashed line denotes the "common" PES obtained by setting θ_T to zeros and the blue shaded region shows

task-specific contributions of SevenNet.MPTrj. Blue and red markers indicate the reference binding energies computed using the corresponding *ab initio* methods (PBE and ω B97M).

Fig. R2. Potential energy surfaces predicted by multi-task SevenNet models trained using different strategies. The first row (**a**, **b**) presents results for models trained on the MPTrj and SPICE databases without DBS, whereas the second row (**c**, **d**) shows the corresponding PES obtained with DBS. The left panels (**a**, **c**) and right panels (**b**, **d**) display results from models trained without and with regularization, respectively.

By comparing Figs. R2a and R2b, we observe that the common PES becomes more physically meaningful after applying regularization, which systematically improves the accuracy of SevenNet.MPTrj while preserving the performance of SevenNet.SPICE. Moreover, this enhanced generalizability of the common PES appears to be independent of the application of DBS and instead primarily attributable to regularization, which also explains the observed synergistic effect between the two approaches. As shown in Fig. R2c, applying DBS improves the accuracy associated with θ_T , however, the common PES remains inaccurate. In contrast, Fig. R2d demonstrates that the most accurate SevenNet.MPTrj is obtained when both θ_T (via DBS) and θ_C (via regularization) are simultaneously improved.

Actions taken:

On page 4, we have added two consecutive paragraphs starting with “We further examine the effects of ...”.

In Supplementary Information, we have added Supplementary Fig. 2.

2) Curriculum Learning Ablation: The training procedure follows a specific curriculum: Crystal → Crystal+Molecule →All. It is unclear if this specific order is necessary or if it introduces bias. Did the authors attempt joint training on all datasets from scratch? An ablation study (or at least a comment) on whether the curriculum order prevents "catastrophic forgetting" of the initial crystal stability would be valuable.

Response:

Before scaling up to the full training set of 15 heterogeneous databases, we conducted a preliminary study for using 4 databases, namely MPtrj, Alex, OMat24 (crystalline systems), and SPICE (molecular systems). When training jointly on all four datasets from scratch, we observed unstable optimization behavior and poor initial convergence. In contrast, initializing the model with parameters trained on the crystalline datasets (MPtrj+Alex+OMat24) and subsequently incorporating the SPICE dataset led to significantly more stable training dynamics and reliable convergence. Similar difficulties in training highly heterogeneous datasets from scratch have also been reported in previous work (doi.org/10.48550/arXiv.2504.21286). Therefore, this curriculum strategy provides a practical and effective solution for achieving stable optimization under limited computational budgets when integrating highly heterogeneous domains, which motivated its adoption in the training of SevenNet-Omni.

Regarding catastrophic forgetting, during training, we also monitored the errors on the previously trained crystal-domain datasets after incorporating additional data (Fig. R3). These errors remained comparable to those obtained prior to the incorporation, indicating that progressively expanding the training dataset does not induce noticeable catastrophic forgetting on previously trained domains. We note, however, that the specific ordering of the curriculum was not rigorously optimized, and a systematic exploration of alternative curriculum orders could be an interesting direction for future work.

Fig. R3. **Prediction accuracies on crystal databases during curriculum learning.** Three models are compared: SevenNet-MF-ompa (7net-ompa), ompa+omol, and SevenNet-Omni (7net-Omni). The ompa+omol denotes an intermediate model which is initialized from 7net-ompa and trained on MPtrj, sAlex, OMat24, and OMol25, along with DBS structures derived from OMol25 (see Section 3.2 in the manuscript). Mean absolute errors (MAEs) of energies, forces, and stress components are evaluated on subsets of each training set. Specifically, the ‘MPA’ set is a subset of MPtrj+Alex database, and the ‘OMat24’ set is a subset of the OMat24 database. As curriculum learning progresses and the material domain of the training set is gradually expanded, the predictive accuracy on crystalline systems remains stable, showing no increase in training errors. The difference in the error balance between 7net-ompa and the other two models originates from the different force loss weights, where 7net-ompa used $\lambda_F = 0.1$ while others used $\lambda_F = 1$.

Actions taken:

On page 5, we have revised sentences starting with “This approach resulted in a more stable learning process, ...”.

In Supplementary Information, we have added Supplementary Fig. 3.

3) Contradiction in DBS Sampling Strategy: There appears to be a contradiction in how the Domain-Bridging Set (DBS) construction is described. In the main text (Page 4), the authors state they “randomly subsample” structures. However, in the Methods section (Section 3.1), they state they “assign higher sampling weights to structures that are (i) structurally distant from inorganic systems...”. This distinction is critical: “random” sampling implies the method works without active selection, whereas “weighted” sampling implies an active learning or adversarial approach is necessary for success. The authors must clarify which method was actually used and, if weighted, how the “structural distance” was quantified.

Response: We thank the reviewer for the critical comment. The statement regarding the main text (Page 4) is about DBS construction in MPtrj + SPICE example related to Fig. 1, while the Methods section (Section 3.1) describes DBS construction for the 7net-Omni. The sampling ratio for each database for DBS in 7net-Omni is determined by assigning a “weight” to each dataset, whereas the selection of individual structures for DBS within a given database is performed uniformly at random. As described in the original manuscript, the sampling ratio is designed to account for two factors: (i) structural dissimilarity from inorganic systems and (ii) differences in computational settings relative to the PBE functional.

To incorporate both aspects, we computed the force MAE of each database using our base model, 7net-ompa. Since 7net-ompa is trained on inorganic structures at the PBE level of theory, the resulting errors reflect contributions from both (a) structural dissimilarity with respect to the 7net-ompa training set and (b) differences between the *ab initio* methods used to label the data and PBE. We did not employ explicit structural distance metrics, advanced similarity measures, or active learning schemes in this procedure.

We agree that clarifying the DBS sampling strategy improves the transparency of the methodology, and we have revised the manuscript accordingly.

Actions taken:

On page 15, we have revised the sentence starting with “For DBS generation ...”.

On page 15, we have revised the two consecutive sentences starting with “This is quantified by the force prediction ...”.

On page 15, we have added the sentence starting with “After assigning total sampling numbers ...”.

4) Clarification of "Performance Improvement": The manuscript frequently uses phrases such as "yields a substantial performance improvement." I recommend the authors be careful with this wording to clearly distinguish between accuracy with respect to the training labels (PBE/DFT) and accuracy with respect to physical reality. Improved performance here implies higher fidelity to the PBE level of theory, but due to the limitations of GGA functionals, this does not necessarily translate to improved accuracy in reality

Response: We agree with the reviewer. We have revised the manuscript to clarify that the term “performance improvement” generally refers to improved agreement with GGA, which does not necessarily imply closer correspondence to physical reality.

Actions taken:

On page 7, we have added two consecutive sentences starting with “We add a cautionary remark that the performance improvements...”.

On page 10, we have revised the sentence starting with “Similar trends were observed for predicting experimental ...”.

5) ORB Benchmarking and Speed: In Figure 5, the inference speed numbers reported for ORB are surprisingly low and appear to disagree with the results reported in the ORB-v3 paper. This discrepancy suggests that the ASE implementation used here may not have been properly optimized. I strongly suggest contacting the ORB developers to help debug this or verify the setup to ensure a fair comparison.

Response:

We thank the reviewer for pointing out this discrepancy. We reviewed our benchmarking procedure and compared with previous studies, and identified important acceleration settings for the ORB models that had been missed in our original setup. Specifically, we enabled `torch.compile` for ORB models and used `cuML` for graph construction. With these changes, we obtained inference speeds consistent with those reported in the ORB-v3 paper (approximately 27 steps/sec for a 1k-atom system). The DPA model results were also updated after correcting an inconsistency in the package configuration used in the previous benchmark.

To further improve the fairness of the comparison, we also lowered the numerical precision where possible, changing the precision of the MACE and NequIP models from float64 to float32. The revised benchmarking results are presented in Fig. R4.

Fig. R4. **Revised inference speed of uMLIPs.** The inference performance of each uMLIP is evaluated using MDs simulations of diamond Si. The speed is reported in units of nanoseconds per day (ns/day), measured on a H100 GPU card.

Actions taken:

On page 14, we have replaced Fig. 5 in the manuscript with new benchmark data.

On page 14, we have revised sentences starting with “The lowest precision settings ...” and added a sentence starting with “For the ORB model, ...”.

On page 14, we have revised the expression “half the speed of MACE” into “one third of the speed of MACE”.

6) Claims regarding ORB Equivariance: The text states: "ORB employs equivariant vector features but does not strictly enforce rotational equivariance throughout the model." I believe this statement may be incorrect. I request the authors double-check the technical details of the specific ORB version used or clarify exactly what is meant by "does not strictly enforce," as this could be misleading.

Response:

We thank the reviewer for raising this point. Equivariant MLIPs typically employ equivariant features, such as vectors and higher-rank tensors, which are often represented using spherical harmonics. To preserve equivariance in the inferred PES, these models apply equivariance-preserving operations, for example Clebsch–Gordan tensor decomposition following tensor products.

ORB models indeed incorporate vector features—specifically, interatomic displacement vectors corresponding to spherical harmonics of degree one—but process them using multi-layer perceptrons rather than explicit tensor operations (see doi.org/10.48550/arXiv.2410.22570 for details of the ORB architecture). This design choice involves a trade-off: while matrix multiplications are generally more computationally efficient than tensor-based operations used in fully equivariant models, they do not strictly preserve rotational invariance of the PES. Consistent with this, the ORB-v3 paper explicitly states that “Contrary to recent literature, we find that non-equivariant, non-conservative architectures can accurately model physical properties, including those which require higher-order derivatives of the potential energy surface” (doi.org/10.48550/arXiv.2504.06231), indicating that strict rotational equivariance is not enforced in ORB models.

To verify the ORB version used in our original manuscript, we computed the PES associated with rigid rotations of an isolated water molecule. Since rigid rotations merely change the reference frame, a strictly rotationally equivariant model should yield invariant energies. As shown in Fig. R5, the ORB[mpa] model exhibits a small amount of energy drift under rigid rotation, whereas the strictly equivariant model (7net-ompa.mpa) remains invariant within numerical precision.

Fig. R5. **Potential energy calculated for water molecules for various uMLIPs.** Input structures for water molecules are generated by a rigidly rotating initial structure with 10° . Blue and red lines indicate energy calculated with ORB[mpa] and 7net-ompa.mpa, respectively.

Rather than enforcing exact equivariance, ORB-v3 adopts a roto-equivariance inducing regularization scheme termed *equigrad*. Although this approach does not guarantee strict equivariance of the PES, it constrains the model such that the predicted energy of a rotated structure does not deviate excessively from that of the original configuration. We therefore believe that our statement regarding the equivariance properties of ORB models is accurate; nevertheless, we agree that clarifying these technical distinctions would improve the readability and completeness of the manuscript.

Actions taken:

On page 5, we have added three consecutive sentences starting with “The architectural categorization of ORB ...”.

7) Figure 1c: The blue solid line should be explicitly defined in the caption

Response: We thank the reviewer for pointing this out and have revised the caption accordingly.

Actions taken:

On page 2, we have revised the caption starting with “Blue and red solid lines represent ...”.

8) Figure 1b: It is not clear what the "task embedding vectors" are or how they were derived.

Response: We appreciate the reviewer for pointing this out. In SevenNet-MF, each task is represented using one-hot encoding; consequently, the task-specific parameters θ_T in each self-interaction layer effectively serve as task-embedding vectors. (doi.org/10.1021/jacs.4c14455) Specifically, the θ_T from the first self-interaction layer of 7net-Omni were used to generate Fig. 1b. To clarify this point, we have added an explicit description of the task-embedding vectors in the revised manuscript.

Actions taken:

On page 5, we have added two consecutive sentences starting with “Since the base architecture, SevenNet-MF, employs ...”.

9) Figure 3c: Please clarify exactly what data set is being used here

Response: The data used in Fig. 3c is identical to the E_{ads} without deformation prediction results shown in Fig. 2f, but is presented as a scattered plot. Specifically, this data is taken from the test set of the GoldDAC dataset (doi.org/10.1016/j.matt.2025.102203), which consists of 26 MOF structures with two types of guest molecules (CO_2 and H_2O). We explicitly clarified that the data in Fig. 3c are the same as those in Fig. 2f to avoid any confusion.

Actions taken:

On page 10, we have added “(X= CO_2 , H_2O)” at the end of the sentence starting with “The adsorption energy, E_{ads} , ...”.

On page 10, we have revised the sentence starting with “The stiffening is further corroborated in Fig. 3c, ...”.

10) Figure 3a: The performance of UMA on OMat seems surprisingly poor. I would double-check this result or clarify if there is a specific reason for this underperformance.

Response:

We appreciate the reviewer’s comment. To check our Linux environment for running UMA, we cross-validated our results using the UMA simulator provided in the official Hugging Face repository (see: https://huggingface.co/spaces/facebook/fairchem_uma_demo). Specifically, we compared the energies of the isolated molecules shown in Fig. 3a after relaxing them in both environments.

As the official UMA simulator currently supports only the UMA-s-1 model (different from the UMA-m-1p1 model benchmarked in our original manuscript) we performed the comparison using UMA-s-1 in our Linux setup as well. Although this model differs from the

one used in the main study, we believe that this procedure is sufficient to demonstrate the validity of our computational environment.

The final energies obtained for each molecule in both environments are presented in Fig. R6. The resulting parity plot demonstrates that our Linux setup reproduces the energies from the official UMA simulator to within numerical precision, confirming the reliability of our computational environment.

Fig. R6. **Scatter plot of molecular energies.** Each value is obtained from the official Hugging Face environment (x-axis) and our Linux setup (y-axis). The gray dashed line shows the perfect match between the two.

While we believe the results reported in our original manuscript are valid, we agree that providing additional discussion on the reason behind relatively poor performance of UMA in Fig. 3a would be helpful to readers.

UMA is a multi-task model trained on OMat24, OMol25, ODAC25, OC20, and OMC25, covering a material and chemical domain comparable in scope to SevenNet-Omni. Nevertheless, its cross-domain generalization is not optimal, as evidenced in Fig. 3a. We attribute this behavior primarily to the following two factors.

First, OMat24 contains a substantially smaller fraction of molecule-containing structures compared to the MPtrj database. We performed an internal analysis aimed at identifying structures with molecule-like characteristics. Specifically, we searched for structures containing H, C, N, O, or F atoms and constructed neighbor graphs based on covalent radii. If more than one disconnected cluster was identified, the structure was classified as a molecule-containing structure. To properly account for single-molecule-in-vacuum cases under periodic boundary conditions, the same analysis was also performed on a $2\times 2\times 2$ supercell.

Using this criterion, we found that approximately 10% of structures in MPtrj correspond to molecule-containing configurations, whereas the fraction is significantly smaller in other datasets, amounting to about 3% in Alex and only 1% in OMat24. As a result, models trained on OMat24 tend to exhibit systematically weaker performance on such molecular structures than models trained on MPtrj-containing datasets, a trend that is also consistently observed in the single-task model performances shown in Fig. 3a.

Second, although UMA (as well as DPA) are multi-task models and include molecular datasets as separate inference tasks, it does not employ task-dependent regularization or domain-bridging sets to facilitate effective knowledge transfer across heterogeneous domains. A similar limitation of the DPA model in low-dimensional benchmarks is also reported in a previous study (doi.org/10.48550/arXiv.2508.15614).

These factors likely lead to reduced robustness when extrapolating to molecular configurations that are weakly represented in the training distribution, resulting in the relatively weaker performance of UMA.omat in Fig. 3a.

Actions taken:

On page 10, we have added sentences starting with “The relatively weaker performance of UMA.omat ...”.

In Supplementary Information, we have added Supplementary Table 2.

Editorial requests regarding checklists

On page 4, we have clarified the performance is evaluated with validation sets in the sentence starting with “All evaluations are performed using ...”.

On page 15, we have added details of the SPICE dataset filtering criteria in the sentence starting with “For the SPICE dataset, ...”.

On page 15, we have added the purpose and details of data splitting in two consecutive sentences starting with “To monitor and evaluate the generalizability toward real-world applications ...”.

Miscellaneous changes

On page 10, we have corrected the reference of the sentence starting with “We suspect that this behavior arises ...”.

On page 17, we have added the acknowledgement of the NVIDIA cuEquivariance team for their support in the cuEquivariance implementation, and revised the project name of “the High-Performance Computing Support Project”.

We have rearranged the indices of the Supplementary Figures and updated the preprint references that were published after the original manuscript submission.